# Compact engineered human mechanosensitive transactivation modules enable potent and versatile synthetic transcriptional control

Barun Mahata[1], Alan Cabrera [1], Daniel A. Brenner[1],
Rosa Selenia Guerra-Resendez [2], Jing Li[1], Jacob Goell[1], Kaiyuan Wang[1],
Yannie Guo [1], Mario Escobar [3], Abinand Krishna Parthasarathy[1],
Hailey Szadowski[2], Guy Bedford[1], Daniel R. Reed[1], Sunghwan Kim[1] &
Isaac B. Hilton [1,2,3] ✉

Engineered transactivation domains (TADs) combined with programmable DNA binding platforms have revolutionized synthetic transcriptional control. Despite recent progress in programmable CRISPR–Cas-based transactivation (CRISPRa) technologies, the TADs used in these systems often contain poorly tolerated elements and/or are prohibitively large for many applications. Here, we defined and optimized minimal TADs built from human mechanosensitive transcription factors. We used these components to construct potent and compact multipartite transactivation modules (MSN, NMS and eN3x9) and to build the CRISPR–dCas9 recruited enhanced activation module (CRISPR-DREAM) platform. We found that CRISPR-DREAM was specific and robust across mammalian cell types, and efficiently stimulated transcription from diverse regulatory loci. We also showed that MSN and NMS were portable across Type I, II and V CRISPR systems, transcription activator-like effectors and zinc finger proteins. Further, as proofs of concept, we used dCas9-NMS to efficiently reprogram human fibroblasts into induced pluripotent stem cells and demonstrated that mechanosensitive transcription factor TADs are efficacious and well tolerated in therapeutically important primary human cell types. Finally, we leveraged the compact and potent features of these engineered TADs to build dual and all-in-one CRISPRa AAV systems. Altogether, these compact human TADs, fusion modules and delivery architectures should be valuable for synthetic transcriptional control in biomedical applications.

Nuclease-deactivated CRISPR–Cas (dCas) systems can be used to modulate transcription in cells and organisms[1–8]. For CRISPR–Cas-based transactivation (CRISPRa) approaches, transcriptional activators can be recruited to genomic regulatory elements using direct fusions to dCas proteins[9–13], antibody-mediated recruitment[14] or using engineered guide RNA (gRNA) architectures[15,16]. High levels of CRISPRa-driven transactivation have been achieved by shuffling[17], reengineering[18] or combining[9,19,20] TADs and/or chromatin modifiers. However, many of the

[1]Department of Bioengineering, Rice University, Houston, TX, USA. [2]Systems, Synthetic, and Physical Biology Graduate Program, Rice University, Houston, TX, USA. [3]Department of BioSciences, Rice University, Houston, TX, USA. ✉e-mail: isaac.hilton@rice.edu

transactivation components used in these CRISPRa systems have coding sizes that are restrictive for applications such as viral vector-based delivery. Moreover, most of the transactivation modules that display high potencies harbor components derived from viral pathogens and can be poorly tolerated in clinically important cell types, embryos and animal models, which could hamper biomedical or in vivo use[21–23]. Finally, there is an untapped repertoire of thousands of human transcription factors (TFs) and chromatin modifiers[24–27] that has yet to be systematically tested and optimized as programmable transactivation components across endogenous target sites, DNA binding platforms and recruitment architectures (for example, direct protein fusions versus aptamer-based). This diverse repertoire of human protein building blocks could be used to reduce the size of transactivation components, obviate the use of viral TFs and possibly permit cell- and/or pathway-specific transactivation.

Mechanosensitive transcription factors (MTFs) modulate transcription in response to mechanical cues and/or external ligands[28,29]. When stimulated, MTFs are shuttled into the nucleus where they can rapidly transactivate target genes by engaging key nuclear factors including RNA polymerase II (RNAP) and/or histone modifiers[30–33]. The dynamic shuttling of MTFs can depend upon both the nature and the intensity of stimulation. Mammalian cells encode several classes of MTFs, including serum-regulated MTFs (for example, YAP, TAZ, SRF, MRTF-A and -B, and MYOCD)[29,34], cytokine-regulated/JAK-STAT family MTFs (for example, STAT proteins)[35] and oxidative stress/antioxidant-regulated MTFs (for example, NRF2)[36], each of which can potently activate transcription when appropriately stimulated. The robust, highly orchestrated and relatively ubiquitous gene regulatory effects of these classes of human MTFs make them excellent potential sources of new nonviral TADs that could be leveraged as components of engineered CRISPRa systems and/or other synthetic gene activation platforms.

Here, we quantify the endogenous transactivation potency of dozens of different TADs derived from human MTFs in different combinations and across various dCas-based recruitment architectures. We use these data to design multipartite transactivation modules, called MSN (MRTF-A/STAT1/eNRF2), NMS (eNRF2/MRTF-A/STAT1) and eN3x9, and we further apply the MSN and NMS effectors to build the CRISPR–dCas9 recruited enhanced activation module (DREAM) platform. We demonstrate that CRISPR-DREAM potently stimulates transcription in primary human cells and cancer cell lines, as well as in murine and CHO cells. We also show that CRISPR-DREAM activates different classes of RNAs spanning diverse regulatory elements within the human genome. Further, we find that the MSN/NMS effectors are portable to smaller engineered dCas9 variants, natural orthologs of dCas9, dCas12a, Type I CRISPR–Cas systems, and transcription activator-like effector (TALE) and ZF proteins. Moreover, we demonstrate that a dCas12a-NMS fusion enables superior multiplexing transactivation capabilities compared with existing systems.

We also show that dCas9-NMS efficiently reprograms human fibroblasts, and we leverage the compact size of these effectors to build potent dual and all-in-one (AIO) CRISPRa AAVs. Finally, we demonstrate that MSN, NMS and eN3x9 are better tolerated than viral-based TADs in primary human mesenchymal stem/stromal cells (MSCs) and T cells. Overall, the engineered transactivation modules that we have developed here are small, highly potent, devoid of viral sequences and versatile across programmable DNA binding systems, and enable robust multiplexed transactivation in human cells—important features that can be leveraged to test new biological hypotheses and engineer complex cellular functions.

## Results

### CRISPR-DREAM displays potent and specific gene activation

We first showed that select TADs[30,37,38] and TAD combinations derived from human MTFs could activate transcription from diverse endogenous human loci when recruited by dCas9 (Supplementary Notes 1 and 2 and Supplementary Figs. 1–8). We leveraged these data to develop the gRNA aptamer/MS2 binding cap protein (MCP)-based recruitment of the MSN and NMS multipartite TAD modules using dCas9. We termed this platform CRISPR-DREAM. To assess the relative transactivation potential of CRISPR-DREAM, we first targeted the DREAM or SAM[15] systems (Fig. 1a,b) to different human promoters in HEK293T cells. All components for both the DREAM and SAM systems were well-expressed in HEK293T cells (Fig. 1c). For all promoters targeted using pools of gRNAs (n = 15), DREAM was superior or comparable to the SAM system (Fig. 1d and Supplementary Fig. 9). Similar results were obtained using antibody staining of CD34 protein levels and flow cytometry analyses in single cells after DREAM or the SAM system (or dCas9-VPR (ref. 9)/dCas9 + MCP-VPR) was targeted to the CD34 promoter (Supplementary Fig. 10). Additionally, when human promoters (n = 11) were targeted using only single gRNAs, DREAM remained superior or comparable to the SAM system in all experiments (Fig. 1e and Supplementary Fig. 11). Interestingly, this trend extended throughout ~1 kilobase (kb) upstream of the transcription start sites surrounding human genes (Supplementary Fig. 12). Collectively, these data demonstrate that, although the DREAM system is smaller than the SAM system, and is devoid of viral TADs, it displays superior or comparable transactivation potency in human cells.

To test the transcriptome-wide specificity of CRISPR-DREAM, we used four gRNAs to target the DREAM or the SAM system to the *HBG1*/*HBG2* locus in HEK293T cells and then performed RNA-sequencing (RNA-seq) (Fig. 1f). *HBG1*/*HBG2* gene activation was specific and potent for both the CRISPR-DREAM and SAM systems relative to dCas9 + MCP-mCherry control-treated cells. However, DREAM activated substantially more H*BG1*/*HBG2* transcription than the SAM system or dCas9-VPR (Fig. 1f and Supplementary Fig. 13). DREAM maintained superior efficacy relative to the SAM system and dCas9-VPR at *HBG1* (and *SBNO2*) across time points (up to at least 12 d) and cell passages (Supplementary Fig. 14). We also found that the DREAM system was significantly (P < 0.05) more potent than the SAM system at all targeted genes when each system was combined with a pool of six gRNAs, each targeting a different gene (Fig. 1g). Additionally, we evaluated the efficacy of the DREAM system across a battery of different human cell types, including a diverse panel of cancer cell lines (Fig. 1h and Supplementary Fig. 15) as well as primary and/or karyotypically normal human cells (Fig. 1i and Supplementary Fig. 16). Finally, we tested the transactivation potency of the DREAM system in mammalian cell types widely used for disease modeling/biocompatibility applications and therapeutic production pipelines (NIH3T3 and CHO-K1 cells, respectively; Supplementary Fig. 17).

From a mechanistic perspective, each of the MRTF-A (M), STAT1 (S) and NRF2 (N) TAD components have been shown to interact with key transcriptional co-factors (Supplementary Fig. 18a). Specifically, individual TADs from MRTF-A, STAT1 and NRF2 can directly interact with endogenous p300 (refs. 32,39), and the Neh4 and Neh5 TADs from NRF2 can also cooperatively recruit endogenous CBP for transcriptional activity[30,40]. Further, MRTF-A and NRF2 can engage other histone modifiers and chromatin remodelers. For example, MRTF-A can complex with JMJD1A, SET1 and BRG1 (refs. 41–43), and NRF2 can also interact with BRG1, as well as CDH6 (refs. 44,45). Moreover, STAT1 and NRF2 can interact with components of the mediator complex[46,47]. Therefore, we suspect that the potency of the engineered MSN and NMS effector proteins is likely related to their robust capacity to recruit powerful and ubiquitous endogenous transcriptional modulators, which is likely positively impacted by their direct tripartite fusion. In support of this hypothesis, we observed that CRISPR-DREAM significantly catalyzed both increased H3K4me3 and H3K27ac at targeted human promoters (Supplementary Fig. 18b,c). Across all experiments the DREAM system displayed highly potent transactivation. However, rare but notable exceptions were targeted endogenous promoters

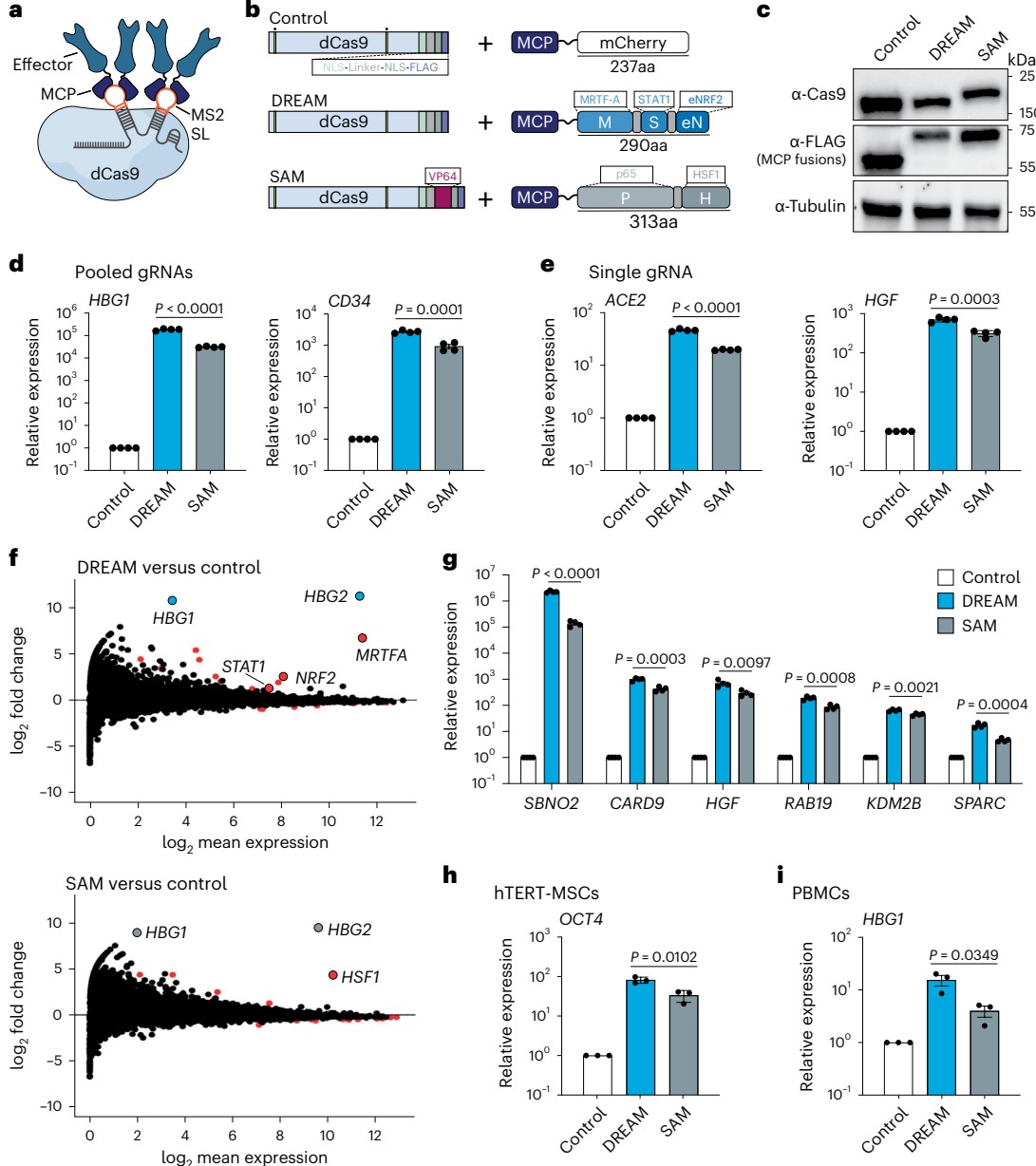

**Fig. 1 | CRISPR-DREAM displays potent activation at human promoters, has high specificity and is robust across cell types. a**, dCas9, a gRNA containing two engineered MS2 stem-loops (MS2 SLs) and MCP-fused transcriptional effector proteins are schematically depicted. **b**, dCas9 and MCP fusion proteins are schematically depicted. Nuclease-inactivating mutations are indicated by yellow bars with dots above. **c**, The expression levels of dCas9 and dCas9-VP64 (top), FLAG-tagged MCP-mCherry, FLAG-tagged MCP-MSN, FLAG-tagged MCP-p65-HSF1 (middle) and β-tubulin (loading control; bottom) are shown as detected by western blotting in HEK293T cells at 72 h post-transfection. **d**,**e**, Relative expression of endogenous human genes after control, DREAM or SAM systems were targeted to their respective promoters using pools of 4 or 3 gRNAs (*HBG1* and *CD34*, respectively; **d**), or using single gRNAs (*ACE2* and *HGF*, respectively; **e**), as measured by qPCR. **f**, RNA-seq data generated after the DREAM or SAM systems were targeted to the *HBG1*/*HBG2* promoter using 4 pooled gRNAs. mRNAs identified as significantly differentially expressed (fold change >2 or

$<$−2 and FDR < 0.05) are shown as red dots. In the top MA plot (CRISPR-DREAM), mRNAs corresponding to *HBG1*/*HBG2* (target genes) are highlighted in light blue. In the bottom MA plot (SAM system), mRNAs corresponding to *HBG1*/*HBG2* (target genes) are highlighted in light gray. mRNAs encoding human components of the MSN or SAM systems shown were also significantly differentially expressed (fold change >2 and FDR < 0.05) and are shown in red. **g**, Six endogenous genes were activated by DREAM or SAM using a pool of gRNAs (1 gRNA per gene) in HEK293T cells. **h**,**i**, *OCT4* (**h**) or *HBG1* (**i**) gene activation by DREAM or SAM systems when corresponding promoters were targeted by 4 gRNAs per promoter in hTERT-MSC or PMBC cells, respectively. All qPCR and RNA-seq samples were processed at 72 h post-transfection. Data are the result of 4 biological replicates for **d**, **e** and **g** and 3 biological replicates for **h** and **i**. See the source data for more information. Data are presented as mean ± s.e.m. *P* values were determined using unpaired two-sided *t*-test. FDR, false discovery rate.

driving highly expressed genes, which were refractory to any synthetic transcriptional activation (Supplementary Fig. 19). Nevertheless, altogether our data demonstrate that CRISPR-DREAM is robust, broadly potent, specific and functionally compatible with diverse human and mammalian cell types.

## CRISPR-DREAM transactivates diverse regulatory elements

Since CRISPR-DREAM efficiently and robustly activated messenger RNAs when targeted to promoter regions, we next tested whether the DREAM system could also activate transcription from distal human regulatory elements (that is, enhancers) and other noncoding transcripts

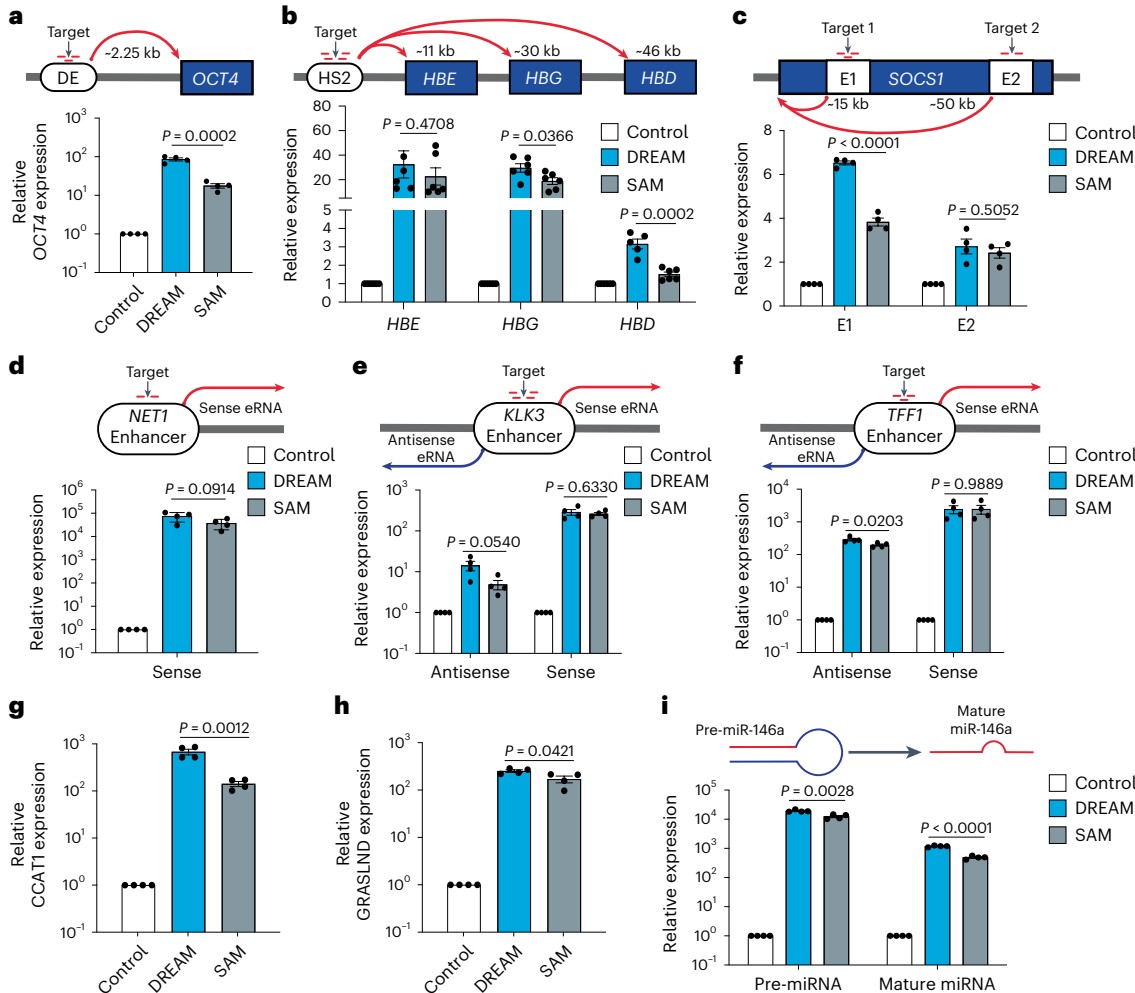

**Fig. 2 | CRISPR-DREAM efficiently activates transcription from diverse human regulatory elements. a–c**, CRISPR-DREAM and the SAM system activated downstream mRNA expression from *OCT4* (**a**); *HBE*, *HBG* and *HBD* (**b**); and *SOCS1* (**c**), when targeted to the *OCT4* distal enhancer (DE), HS2 enhancer or one of two intragenic *SOCS1* enhancers, using pools of 3 (*OCT4* DE), 4 (*HS2*), 3 (*SOCS1* + 15 kb) or 2 (*SOCS1* + 50 kb) gRNAs, respectively. **d**, CRISPR-DREAM and the SAM system activated sense eRNA expression when targeted to the *NET1* enhancer using 2 gRNAs. **e,f**, CRISPR-DREAM and the SAM system bidirectionally activated eRNA expression when targeted to the *KLK3* (**e**) or *TFF1* (**f**) enhancers

using pools of 4 or 3 gRNAs, respectively. **g,h**, CRISPR-DREAM and the SAM system activated the expression of lncRNA when targeted to the CCAT1 (**g**) or GRASLND (**h**) promoters using pools of 4 gRNAs, respectively. **i**, CRISPR-DREAM and the SAM system activated the expression of pre- and mature miR-146a when targeted to the miR-146a promoter using a pool of 4 gRNAs. All samples were processed for qPCR at 72 h post-transfection. Data are the result of 5 or 6 biological replicates for **b** and 4 biological replicates for **a** and **c**–**i**. See the source data for more information. Data are presented as mean ± s.e.m. *P* values were determined using unpaired two-sided *t*-test. E1, Enhancer 1; E2, Enhancer 2.

(that is, enhancer RNAs (eRNAs), long noncoding RNAs (lncRNAs) and microRNAs (miRNAs)). We first targeted the DREAM or SAM system to the *OCT4* distal enhancer[48] and found that the DREAM system significantly (*P* < 0.05) upregulated *OCT4* expression relative to the SAM system when targeted to the distal enhancer (Fig. 2a). Similar results were observed when targeting the DREAM system to the DRR enhancer[49] upstream of the *MYOD* gene (Supplementary Fig. 20a). We also targeted the DREAM system to the human HS2 enhancer[50,51] and observed that the DREAM system induced expression from the downstream *HBE*, *HBG* and *HBD* genes (Fig. 2b). We further observed transactivation of the *SOCS1* gene when the DREAM system was targeted to either of two different intragenic *SOCS1* enhancers: one located ~15 kb and the other ~50 kb downstream of the *SOCS1* transcription start site (Fig. 2c). Together, these data demonstrate that CRISPR-DREAM can stimulate human gene expression when targeted to different classes of enhancers (those regulating a single-gene or multiple genes, or intragenic enhancers) embedded within native chromatin.

We next tested whether CRISPR-DREAM could activate eRNAs when targeted to endogenous human enhancers. When targeted to

the *NET1* enhancer, the DREAM system activated eRNA transcription (Fig. 2d), consistent with other reports[52]. Moreover, when the DREAM system was targeted to the bidirectionally transcribed *KLK3* and *TFF1* enhancers, we observed substantial upregulation of eRNAs in both the sense and antisense directions (Fig. 2e,f). Similar results were obtained when targeting the human *FKBP5* and *GREB1* enhancers (Supplementary Fig. 20b,c). CRISPR-DREAM also stimulated the production of endogenous lncRNAs when targeted to the CCAT1, GRASLND, HOTAIR or MALAT1 loci (Fig. 2g,h and Supplementary Fig. 20d,e). Finally, we found that the DREAM system activated miRNA-146a expression when targeted to the miRNA-146a promoter (Fig. 2i). Taken together, these data show that CRISPR-DREAM can robustly transactivate regulatory regions spanning diverse classes of the human transcriptome.

## Orthogonal CRISPR-DREAM platforms expand genomic targeting

To enhance the versatility of CRISPR-DREAM beyond SpdCas9 and to expand targeting to non-NGG PAM sites, we selected the two smallest naturally occurring orthogonal Cas9 proteins, SadCas9

(1,096 amino acids (aa)) and CjdCas9 (1,027aa), for further analyses (Fig. 3a,d). We used SadCas9-specific gRNAs harboring MS2 loops[53] to compare the potency between the SadCas9-DREAM and SAM systems in HEK293T cells. SadCas9-DREAM was significantly ($P < 0.05$) more potent than SadCas9-SAM when targeted to either the *HBG1* or *TTN* promoter (Fig. 3b). We also found that SadCas9-DREAM outperformed or was comparable to SadCas9-VPR when targeted to these loci (Fig. 3c) and that SadCas9-DREAM displayed high levels of transactivation in a second human cell line (Supplementary Fig. 21a,b).

CjdCas9-based transcriptional activation platforms have also recently been developed using viral TADs (for example, miniCAFE)[54]; however, gRNA-based recruitment of transcriptional modulators using CjdCas9 has not been described. Therefore, we engineered the CjdCas9 gRNA scaffold to incorporate an MS2 loop within the tetraloop of the CjCas9 gRNA scaffold (Supplementary Fig. 21c). We used this MS2-modified CjdCas9 gRNA to generate CjdCas9-DREAM and compared the potency between CjdCas9-DREAM, CjdCas9-SAM and CjdCas9-VPR at the *HBG1* or *TTN* promoters (Fig. 3e,f) in HEK293T cells. At all targeted sites, CjdCas9-DREAM outperformed or was comparable to the CjdCas9-SAM or CjdCas9-VPR systems. We also observed high levels of transactivation using CjdCas9-DREAM in a different human cell line (Supplementary Fig. 21d,e). These data demonstrate that DREAM is not only compatible to other orthogonal dCas9 targeting systems, but that it displays superior performance in terms of CRISPRa activity at most tested promoters.

## Generation and validation of a compact mini-DREAM system

We next sought to reduce the sizes of the CRISPR-DREAM components. We first investigated whether individual TADs could be minimized while still retaining the transactivation potency when recruited by dCas9. We focused on individual TADs from MTFs that displayed transactivation potential (that is, MRTF-A, MRTF-B and MYOCD proteins; Supplementary Figs. 1 and 2). As mentioned above, 9aa TADs have been shown to synthetically activate transcription previously using GAL4 systems[55,56]. Therefore, we used predictive software[55] to identify 9aa TADs in MRTF-A, MRTF-B and MYOCD proteins, and recruited these TADs to human loci using dCas9 and MCP-MS2 fusions in single, bipartite and tripartite formats (Supplementary Note 3 and Supplementary Fig. 22). Interestingly, we observed that while single 9aa TADs did not activate endogenous gene expression, tripartite combinations of 9aa TADs were able to robustly activate endogenous genes, albeit to varying degrees (Supplementary Fig. 22f). We selected one tripartite 9aa combination (3x 9aa TAD; MRTF-B.3 + MYOCD.1 + MYOCD.3) for further analysis (Fig. 3g). This 3x 9aa TAD activated *HBG1*, *TTN* and *CD34* gene expression when recruited to corresponding promoters using dCas9 (Fig. 3h and Supplementary Fig. 22g). We also found that this 3x 9aa

TAD combination could activate gene expression via a single gRNA, and moreover could transactivate other endogenous regulatory loci (Supplementary Fig. 22h–j). These results suggest that combinations of 9aa TADs can be used as minimal functional units to transactivate endogenous human loci when recruited via dCas9.

We next combined the 3x 9aa TAD with the engineered NRF2 TAD (eNRF2) in four different combinations to generate a small yet potent transactivation module called eN3x9 (Supplementary Fig. 23). Notably, minimized Cas9 proteins that retain DNA binding activity have also been recently created[57,58]. Therefore, we next evaluated the relative transactivation capabilities among a panel of minimized, HNH-deleted dCas9 variants in tandem with MCP-MSN and found that an HNH-deleted variant without a linker between two RuvC domains was optimal, albeit with slight protein expression decreases (Supplementary Fig. 24a,b). We further validated this linker-less, HNH-deleted CRISPR-DREAM variant at multiple human promoters and other regulatory elements (Supplementary Fig. 24c–h) and then combined this minimized dCas9 with MCP-eN3x9 to generate the mini-DREAM system (Fig. 3i). The mini-DREAM system transactivated *HBG1*, *TTN* and *IL1RN* gene expression when recruited to corresponding promoters (Fig. 3j and Supplementary Fig. 25a). We also found that the mini-DREAM system could activate endogenous promoters via a single gRNA (Supplementary Fig. 25b,c) and could activate downstream gene expression when targeted to an upstream enhancer (Supplementary Fig. 25d).

To demonstrate the utility of the mini-DREAM platform, we used this system to create progesterone-producing HEK293T cell factories. Specifically, we simultaneously targeted and activated three key genes in the progesterone production pathway (*STAR*, *CYP11A1* and *HSD3B2*), which resulted in increased target gene expression and significant production of progesterone (Fig. 3k–n). We also evaluated whether the minimized components of the mini-DREAM system were functional when delivered within a single vector (Supplementary Fig. 25e) and found that this compact, single-vector mini-DREAM system retained transactivation potential when targeted to human promoters using pooled gRNAs or a single gRNA (Supplementary Fig. 25f–i). Notably, mini-DREAM and mini-DREAM compact also outperformed the miniCAFE platform at two different loci in HEK293T cells (Supplementary Fig. 25j,k). Overall, these data show that the components of the CRISPR-DREAM system can be substantially minimized while retaining functionality.

## The MSN and NMS effectors are robust and versatile

We next tested the potency of tripartite MSN and NMS effectors when fused to dCas9 in different architectures and observed that both effectors could activate gene expression when fused to the N or C terminus

**Fig. 3 | CRISPR-DREAM is portable to orthogonal dCas9 proteins and amenable to miniaturization. a**, The SadCas9-DREAM system is schematically depicted. Nuclease-inactivating mutations are indicated by yellow bars with dots above. **b**, *HBG1* or *TTN* gene activation using the SadCas9-DREAM or SadCas9-SAM system, when recruited using pools of 4 promoter-targeting gRNAs. **c**, *HBG1* or *TTN* gene activation using the SadCas9-DREAM or SadCas9-VPR system, when recruited using pools of 4 MS2-modifed (SadCas9-DREAM) or standard promoter-targeting gRNAs (SadCas9-VPR), respectively. **d**, The CjdCas9-DREAM system is schematically depicted. Nuclease-inactivating mutations are indicated by yellow bars with dots above. **e**, *HBG1* or *TTN* gene activation using the CjdCas9-DREAM or CjdCas9-SAM system, when recruited using pools of 3 MS2-modified promoter-targeting gRNAs. **f**, *HBG1* or *TTN* gene activation using the CjdCas9-DREAM or CjdCas9-VPR system, when recruited using pools of 3 MS2-modifed (SadCas9-DREAM) or standard promoter-targeting gRNAs (CjdCas9-VPR), respectively. **g**, A 3x 9aa TAD derived from MYOCD and MRTF-B TADs is schematically depicted; GS, glycine-serine linker. **h**, *HBG1* or *TTN* gene activation when the 3x 9aa TAD was fused to MCP and recruited using dCas9 and a pool of 4 MS2-modified promoter-targeting gRNAs. **i**, The mini-DREAM

system is schematically depicted. MCP-eN3x9 is a fusion protein consisting of MCP, eNRF2 and the 3x 9aa TAD from **g**. **j**, *HBG1* or *TTN* gene activation when either the mini-DREAM or CRISPR-DREAM system was recruited using a pool of 4 MS2-modified promoter-targeting gRNAs. **k**, A simplified biosynthetic pathway for progesterone production is schematically depicted. **l**, The workflow to build progesterone-producing HEK293T cell factories using the mini-DREAM platform and corresponding gRNA array is shown. **m**, *STAR*, *CYP11A1* and *HSD3B2* gene activation after mini-DREAM-transduced HEK293T cells were transfected with the indicated gRNA array or a nontargeting gRNA control plasmid. **n**, Secreted progesterone levels after mini-DREAM-transduced HEK293T cells were transfected with the indicated gRNA array or a nontargeting gRNA control plasmid. All samples were processed for qPCR or ELISA at 72 h post-transfection. Data are the result of 4 biological replicates for **b**, **c**, **e**, **f**, **j**, **m** and **n**, and 3 or 4 biological replicates for **h**. See the source data for more information. Data are presented as mean ± s.e.m. *P* values were determined using unpaired two-sided *t*-test. BH, bridge helix; eN, engineered NRF2; M, MRTF-A; PI, PAM-interacting domain; REC, recognition lobe; S, STAT1.

of dCas9 (Supplementary Note 4 and Supplementary Fig. 26) or when recruited via the SunTag[14] architecture (Supplementary Fig. 27). Interestingly, in contrast to MCP-mediated recruitment (Supplementary Fig. 8), additional TADs were observed to improve performance in direct fusion architectures (Supplementary Fig. 26a and Supplementary Note 4). In the SunTag architecture, the NMS domain was superior to other benchmarked effector domains, such as VP64 (ref. 14), VPR[59] and p65-HSF1 (ref. 60) (Supplementary Fig. 27a–c). To maximize the

potential use of the MSN/NMS effector domains and explore their versatility, we next tested whether each was capable of gene activation when fused to TALE or ZF scaffolds (Fig. 4a,b). Both effectors strongly transactivated *IL1RN* using a single TALE fusion protein (Supplementary Fig. 28) or a pool of four TALE fusion proteins targeted to the *IL1RN* promoter (Fig. 4a). Similarly, both effectors activated *ICAM1* expression using a single synthetic ZF fusion protein targeted to the *ICAM1* promoter (Fig. 4b). These data demonstrate that the MSN and NMS

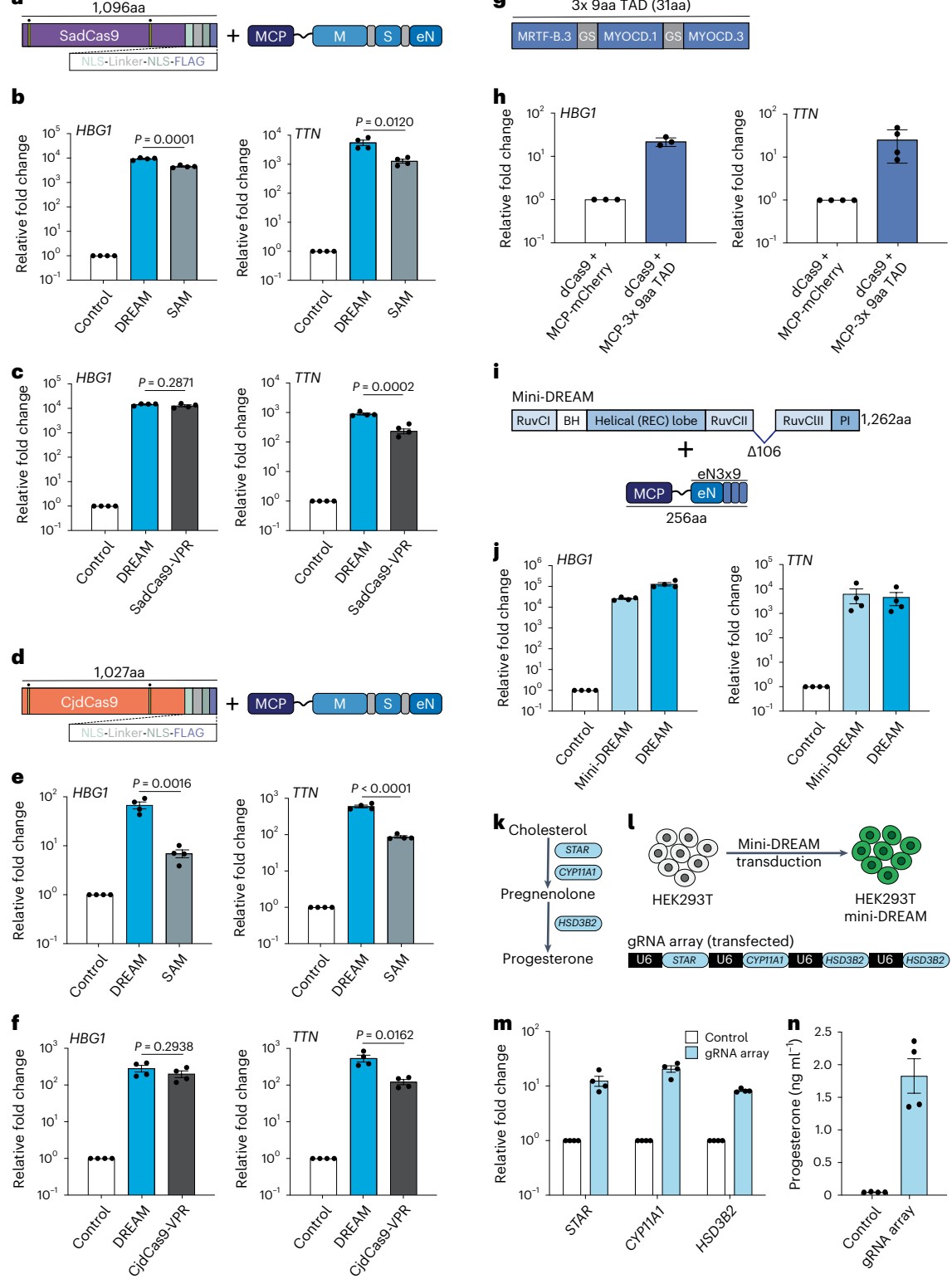

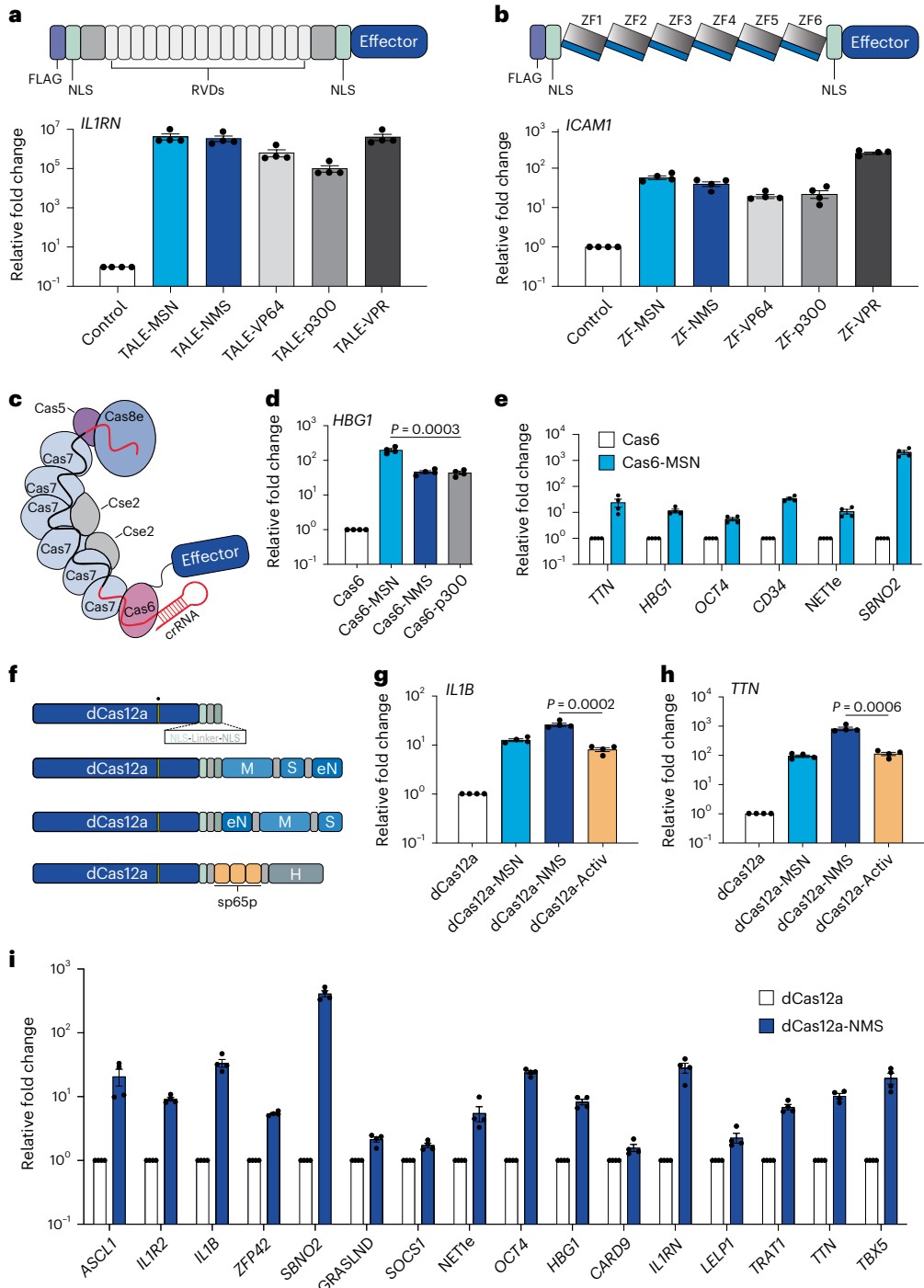

**Fig. 4 | The MSN and NMS effector domains are portable to diverse DNA binding platforms and enable superior multiplexing when fused to dCas12a.** **a**, Synthetic TALE proteins harboring indicated effector domains were designed to target the human *IL1RN* promoter. RVD, repeat variable di-residue. Relative *IL1RN* expression (bottom) at 72 h after indicated TALE fusion protein-encoding plasmids were transfected. **b**, Synthetic zinc finger (ZF) proteins harboring indicated effector domains were designed to target the human *ICAM1* promoter. Relative *ICAM1* expression (bottom) at 72 h after indicated ZF fusion protein-encoding plasmids were transfected. **c**, The Type I CRISPR system derived from *E. Coli* K-12 (Eco-Cascade) is schematically depicted along with an effector fused to the Cas6 protein subunit. **d**, *HBG1* gene activation when the MSN, NMS or p300 effector domains were fused to Cas6 and the respective engineered Eco-Cascade complexes were targeted to the *HBG1* promoter using a single crRNA. **e**, Multiplexed activation of 6 endogenous genes at 72 h after co-transfection

of Eco-Cascade complexes when MSN was fused to Cas6 and targeted using a single crRNA array expression plasmid (1 crRNA per promoter). **f**, The dCas12a protein and indicated fusions are schematically depicted along with the E993A DNase-inactivating mutation indicated by a yellow bar with a dot above. **g,h**, *IL1B* (**g**) or *TTN* (**h**) gene activation using the indicated dCas12a fusion proteins when targeted to each corresponding promoter using a pool of 2 crRNAs (for *IL1B*) or a single array encoding 3 crRNAs (*TTN*), respectively. **i**, Multiplexed activation of 16 indicated endogenous genes at 72 h after co-transfection of dCas12a-NMS and a single crRNA array expression plasmid encoding 20 crRNAs. All samples were processed for qPCR at 72 h post-transfection in HEK293T cells. See the source data for more information. Data are the result of 4 biological replicates for **a**, **b**, **d**, **e**, **g**, **h** and **i**. Data are presented as mean ± s.e.m. *P* values were determined using unpaired two-sided *t*-test. NLS, nuclear localization signal.

effectors are compatible with diverse programmable DNA binding scaffolds beyond CRISPR–Cas systems.

Transcriptional activators have recently been shown to modulate the expression of endogenous human loci when recruited by Type I CRISPR systems[61]. Therefore, to evaluate whether MSN and/or NMS were functional beyond Type II CRISPR systems, we fused each to the Cas6 component of the *Escherichia coli* Type I CRISPR Cascade (Eco-Cascade) system (Fig. 4c). Our data showed that in most cases Cas6-MSN (or NMS) performed better than the Cas6-p300 system when targeted to a spectrum of human promoters (Fig. 4d and Supplementary Fig. 29a–d). We also observed that the Cas6-MSN (or NMS) systems could activate eRNAs when targeted to the endogenous *NET1* enhancer (Supplementary Fig. 29e). One advantage of CRISPR–Cascade is that the system can process its own CRISPR RNA (crRNA) arrays, which can enable multiplexed targeting to the human genome. Previous reports have leveraged this capability to simultaneously activate two human genes[61]. We found that when Cas6 was fused to MSN, the CRISPR–Cascade system could simultaneously activate up to six human genes when corresponding crRNAs were co-delivered in an arrayed format (Fig. 4e and Supplementary Fig. 29f). We also found that these transactivation capabilities were extensible to another Type I CRISPR system, *Pae*-Cascade[62] (Supplementary Fig. 29g–i) and that the NMS effector enabled superior multiplexed gene activation when fused to dCas12a, a Type V CRISPR system (Fig. 4g–i, Supplementary Note 5 and Supplementary Fig. 30a–e). In sum, these data show that the MSN and NMS effectors are robust and directly compatible with programmable DNA binding platforms beyond Type II CRISPR systems without any additional engineering.

## Efficient reprogramming of human fibroblasts using dCas9-NMS

CRISPRa systems using repeated portions of the alpha herpesvirus VP16 TAD (dCas9-VP192) have been used to efficiently reprogram human foreskin fibroblasts (HFFs) into induced pluripotent stem cells (iPSCs)[17]. To evaluate the functional capabilities of our engineered human transactivation modules, we fused the NMS domain directly to the C terminus of dCas9 (dCas9-NMS) and tested its ability to reprogram HFFs. We used a direct dCas9 fusion architecture so that we could leverage gRNAs previously optimized for this reprogramming strategy and to better compare dCas9-NMS with the corresponding state of the art (dCas9-VP192)[17]. We used the NMS effector as opposed to MSN, as NMS displayed more potency than MSN when directly fused to dCas9 (Supplementary Fig. 26a). We targeted dCas9-NMS (or dCas9-VP192) to endogenous loci using the 15 gRNAs previously optimized to reprogram HFFs to pluripotency with the dCas9-VP192 system. Using this approach, we observed morphological changes beginning by 8 d post-nucleofection (Fig. 5a) and efficient reprogramming by 16 d post-nucleofection, although to a lesser extent than when using dCas9-VP192 (Supplementary Fig. 31a).

We picked and expanded iPSC colonies and then measured the expression of pluripotency and mesenchymal genes ~40 d post-nucleofection. We found that genes typically associated with pluripotency (*OCT4*, *SOX2*, *NANOG*, *LIN28A*, *REX1*, *CDH1* and *FGF4*)[63,64] were highly expressed in colonies derived from HFFs nucleofected with the gRNA cocktail and dCas9-NMS or dCas-VP192 (Fig. 5b and Supplementary Fig. 31b–f). Conversely, we observed that genes typically associated with fibroblast/mesenchymal cell identity (*THY1*, *ZEB1*, *ZEB2*, *TWIST* and *SNAIL2*)[63,64] were poorly expressed in colonies derived from HFFs nucleofected with the gRNA cocktail and dCas9-NMS or dCas-VP192 (Fig. 5c and Supplementary Fig. 31g–i). Finally, we assessed the expression of pluripotency-associated markers (SSEA-4, TRA-1-81 and TRA-1-60)[65], and found that all were highly expressed in iPSC colonies derived from HFFs nucleofected with the gRNA cocktail and either dCas9-NMS or dCas-VP192 (Fig. 5d,e and Supplementary Fig. 31j). These data show that engineered transactivation modules

sourced from human MTFs can be used to efficiently reprogram complex cell phenotypes, including cell lineage.

## Engineered TADs are well tolerated in primary human cells

The recent development of CRISPRa tools has enabled new therapeutic opportunities[6,66]. However, it has been shown that in some cases, CRISPRa tools harboring viral TADs can be poorly tolerated, and even toxic[12,21–23]. This prompted us to test the relative expression and efficacy of the human MTF-derived multipartite TAD MSN, NMS and eN3x9 tools in comparison with the viral multipartite TAD VPR in therapeutically relevant human primary cells. We selected primary human umbilical cord MSCs and primary T cells for analysis. Lentiviral transduction was selected to ensure high levels of payload delivery. Interestingly, we observed that lentiviral titers were influenced by fused TADs, with MCP fused to eN3x9 consistently generating the highest titers (Supplementary Fig. 32). We next transduced MSCs using a multiplicity of infection (MOI) of ~10.0 for all conditions and observed variable expression levels among MCP fusion proteins at 72 h post-transduction using both microscopy and flow cytometry (Fig. 6a), despite using equal amounts of lentivirus. For instance, although MCP-eN3x9 and MCP-NMS displayed high levels of expression via microscopy, MCP-VPR and MCP-MSN were relatively poorly expressed. Similarly, we tested the expression levels of these MCP fusions in primary T cells using lentiviral transduction at a fixed MOI of ~5.0 across conditions and observed that MCP-eN3x9 displayed the highest expression levels at 72 h post-transduction, while MCP-VPR showed the lowest expression (Fig. 6b).

We next assessed the gene activation capabilities of these MCP–TAD fusions in primary MSCs and T cells. In MSCs, eN3x9 outperformed all other effectors, and VPR showed the lowest potency when targeted to the *TTN* promoter (Fig. 6c). In primary T cells each TAD activated *CARD9* expression to relatively similar and modest levels when targeted to the *CARD9* promoter (Fig. 6d). However, in primary T cells we observed that the human MTF-derived multipartite TADs resulted in dramatically better T cell viability than the viral multipartite TAD VPR (Supplementary Fig. 33). Interestingly, these effects were less obvious in transformed cell lines (Supplementary Fig. 34). Collectively these data demonstrate that the human MTF-derived multipartite MSN, NMS and eN3x9 TADs are as potent as or more potent than the VPR TAD, while also maintaining similar or superior expression levels in therapeutically relevant human primary cells. Notably, MSN, NMS and eN3x9 are also much smaller than the VPR TAD, and in the case of primary T cells are also much less cytotoxic.

## Streamlined AAV-mediated delivery of CRISPR-DREAM components

AAV-mediated delivery has emerged as a powerful method to deliver therapeutic payloads in vitro[67] and in vivo[68]. However, due to strict payload limitations, the delivery of CRISPRa tools using AAV has been limited to dual AAV systems and/or the use of viral TADs[69,70]. To assess the transcriptional activation potential of the compact CRISPR-DREAM components in combination with AAV-mediated delivery, we targeted the murine *Agrp* gene, which modulates food intake behavior and obesity[71,72], as a proof of concept. We first tested 15 individual gRNAs targeting a ~1-kb window upstream of the *Agrp* promoter in Neuro-2a cells to identify a top performing gRNA (Supplementary Fig. 35a,b). Based on these results, we constructed a dual AAV delivery system, wherein one AAV expressed dCas9, and the other AAV expressed the top performing *Agrp*-targeting gRNA, along with MCP-MSN (Fig. 6e and Supplementary Fig. 35c). Both recombinant AAVs (and an EGFP control AAV) used the AAV8 serotype capsid to ensure efficient neuronal transduction[73] (Supplementary Fig. 35d). In dual AAV-transduced (dCas9 and gRNA/MCP-MSN, respectively) primary murine neurons, we observed high levels of *Agrp* activation (Fig. 6f).

Encouraged by this result using a dual AAV strategy, we next designed two different all-in-one (AIO) AAV approaches (Fig. 6g). These

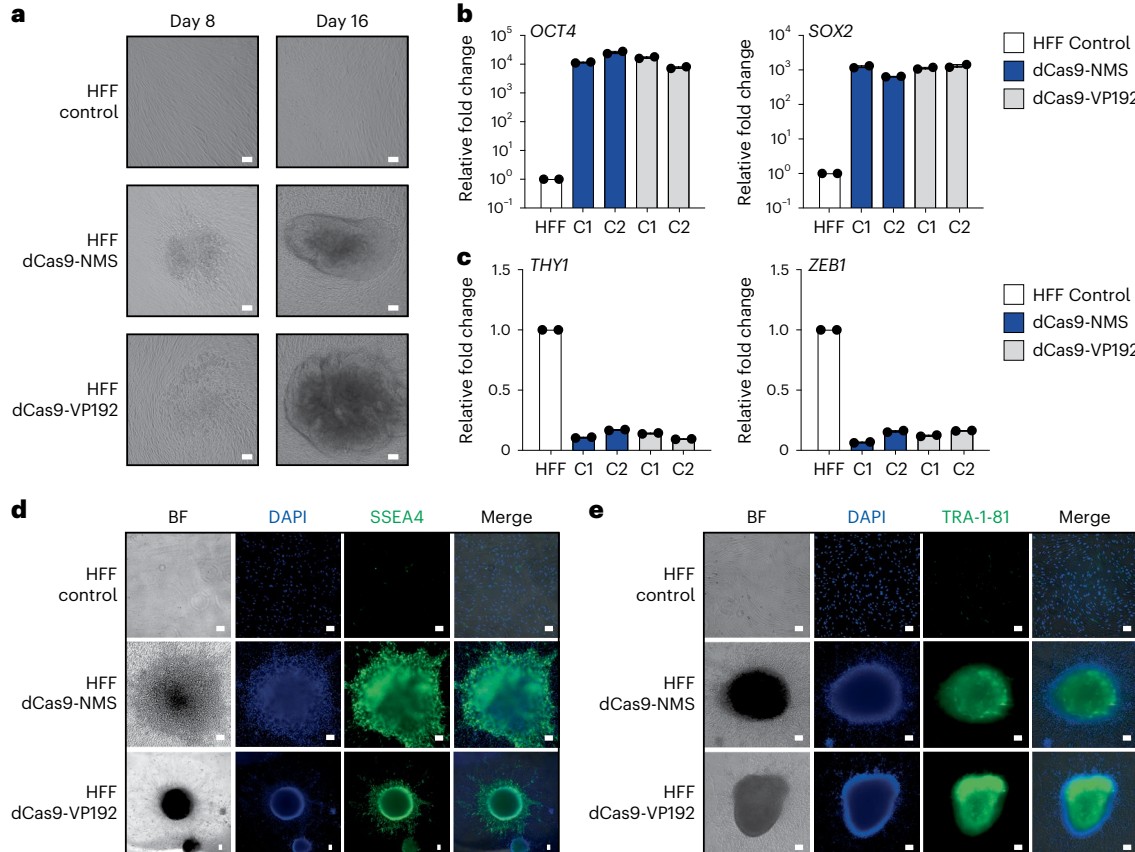

**Fig. 5 | dCas9-NMS permits efficient in vitro reprogramming of human fibroblasts. a**, Primary HFFs were nucleofected with plasmids encoding 15 multiplexed gRNAs targeting the *OCT4*, *SOX2*, *KLF4*, c-*MYC* and *LIN28A* promoter and EEA motifs (as previously reported[17]), and either dCas9-NMS (middle row) or dCas9-VP192 (bottom row). HFF morphology was analyzed 8 and 16 d later (white scale bars, 100 μm). **b**, Relative expression of pluripotency-associated genes *OCT4* (left) and *SOX2* (right) in representative iPSC colonies (C1 or C2) approximately 40 d after nucleofection of either dCas9-NMS (blue) or dCas9-VP192 (gray) and multiplexed gRNAs compared with untreated HFF controls. *n* = 2 independent measurements from independent subclones per colony. **c**, Relative expression of mesenchymal-associated genes *THY1* (left) and *ZEB1* (right) in representative iPSC colonies (C1 or C2) approximately 40 d after

nucleofection of either dCas9-NMS (blue) or dCas9-VP192 (gray) and multiplexed gRNAs compared with untreated HFF controls. *n* = 2 independent measurements from independent subclones per colony. **d,e**, Immunofluorescence microscopy of HFFs approximately 40 d after nucleofection of either dCas9-NMS or dCas9-VP192 and multiplexed gRNAs compared with untreated HFF controls (white scale bars, 100 μm). Cells were stained for the expression of pluripotency-associated cell surface markers SSEA4 (**d**, green) or TRA-1-81 (**e**, green). All cells were counterstained with DAPI for nuclear visualization. Data presented in **a** is a representative of 3 independent experiments. Data are the result of 2 biological independent measurements from independent subclones per colony for **b** and **c**. Data presented in **d** and **e** are representative of 2 independent experiments. Data are presented as mean ± s.e.m.

designs leveraged the M11 promoter to express a gRNA, and either the SCP1 or EFS promoter to drive the expression of NMS fused to the N terminus of SadCas9. NMS was prioritized over MSN as it showed higher potency when fused to the N terminus of dCas9 (Supplementary Fig. 24). To further reduce packaging size, we also selected compact engineered WPRE and polyA[74] tail elements in these construct designs. After selecting a top performing *Agrp*-targeting SadCas9 gRNA in Neuro-2A cells (Supplementary Fig. 35e,f), we made recombinant AAVs (using serotype AAV8) and delivered these AIO AAVs to primary murine neurons. In both cases, we observed significant (*P* < 0.05) transcriptional upregulation of *Agrp*, with the EFS promoter-harboring vector displaying superiority to the SCP promoter-harboring vector (Fig. 6h). These data demonstrate that the compact components of CRISPR-DREAM retain high transactivation potency when delivered into primary cells using either dual or AIO AAV modalities.

## Discussion

Here, we harnessed the programmability and versatility of different dCas9-based recruitment architectures (direct fusion, gRNA aptamer and SunTag-based) to optimize the transcriptional output of TADs

derived from natural human TFs. We leveraged these insights to build superior and widely applicable transactivation modules that are portable across all modern synthetic DNA binding platforms, and that can activate the expression of diverse classes of endogenous RNAs. We selected MTFs for biomolecular building blocks because they naturally display rapid and potent gene activation at target loci and can interact with diverse transcriptional co-factors across different human cell types, and because their corresponding TADs are relatively small[75–77]. We not only identified and validated the transactivation potential of TADs sourced from individual MTFs, but we also established the optimal TAD sequence compositions and combinations for use across different synthetic DNA binding platforms, including Type I, II and V CRISPR systems, TALE proteins and ZF proteins.

Additionally, our study demonstrated that the superior transactivation capabilities of the CRISPR-DREAM system, again consisting of dCas9 and a gRNA aptamer-recruited MCP-MSN fusion, are not reliant upon the direct fusion(s) of any other proteins (viral or otherwise) to dCas9, in contrast to the SAM system which relies upon dCas9-VP64 (ref. 15). We also integrated the MSN and NMS effectors with the Type I CRISPR–Cas and Type II dCas12a platforms to enable superior

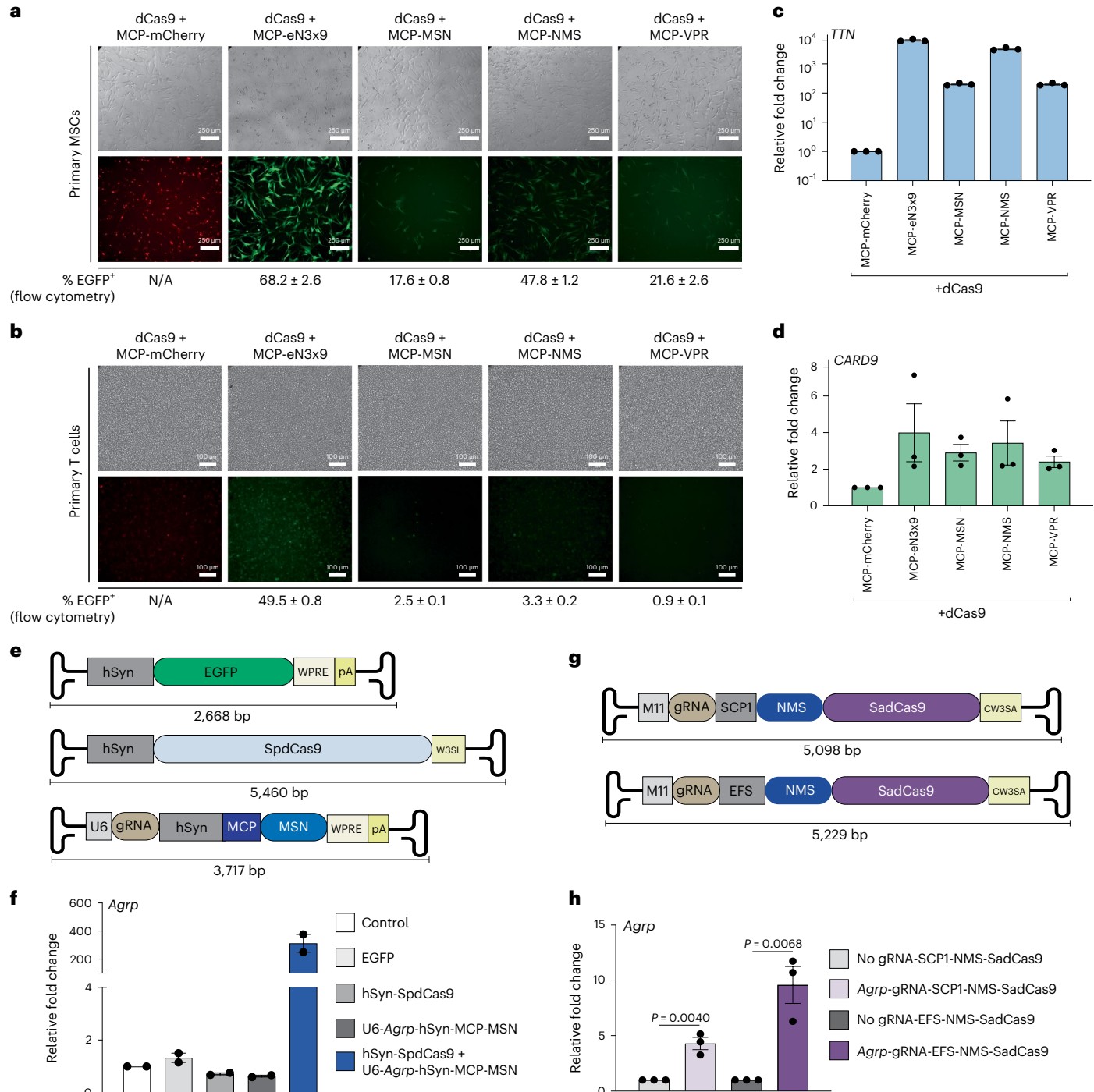

**Fig. 6 | CRISPR-DREAM components are well tolerated in primary cells and compatible with viral delivery methods. a,b,** Immunofluorescence microscopy showing mCherry/EGFP expression levels in MSCs (**a**) and human T cells (**b**) at 72 h after co-transduction of dCas9 in combination with either MCP-mCherry (control), MCP-eN3x9-T2A-EGFP, MCP-MSN-T2A-EGFP, MCP-NMS-T2A-EGFP or MCP-VPR-T2A-EGFP, respectively (white scale bars, 250 μm for MSCs; 100 μm for T cells). MCP-fusion vectors also contain a U6-driven gRNA expression cassette and either a *TTN* (MSCs) or *CARD9* (T cells). **c,d,** Relative expression of *TTN* (**c**) or *CARD9* (**d**) in MSCs and T cells, respectively, 3 d after lentiviral co-transduction using indicated components. **e,** AAV constructs used for dual-delivery of CRISPR-DREAM components are schematically depicted. The EFGP control vector is shown (top) along with the hSyn promoter-driven SpdCas9 vector (middle), which consists of a modified WPRE/polyA sequence (W3SL). The U6

promoter-driven gRNA expressing vector (bottom) is also shown and also encodes MCP fused to MSN, which is driven by the hSyn promoter. **f,** *Agrp* gene activation in mouse primary cortical neurons using the dual AAV8 transduced CRISPR-DREAM system described (in **e**) at 5 d post-transduction. **g,** AIO SadCas9-based AAV vectors are schematically depicted. AIO vectors consist of M11 promoter-driven gRNA cassettes and either SCP1 (top) or EFS (bottom) promoter-driven NMS-SadCas9. A modified WPRE/polyA sequence (CW3SA) was used in the AIO vectors. **h,** *Agrp* gene activation in mouse primary cortical neurons transduced with AIO AAV vectors (in **h**) at 5 d post-transduction. Data are the result of 3 biological replicates for **c, d** and **h** and 3 biological replicates for **f**. See the source data for more information. Data are presented as mean ± s.e.m. *P* values were determined using unpaired two-sided *t*-test. N/A, not applicable.

multiplexed endogenous activation of human genes. This multiplexing capability holds tremendous promise for reshaping endogenous cellular pathways and/or engineering complex transcriptional networks. dCas9-based TFs harboring viral TADs have also been used for directed differentiation and cellular reprogramming[9,17,78,79]. Here, we showed that we could reprogram human fibroblasts into iPSCs using dCas9 directly fused to the NMS transcriptional effector using established protocols[17]. Further, we demonstrated that the MSN and NMS effectors were compatible with dual and AIO AAV vectors, which empower researchers with a streamlined modality to induce endogenous gene expression in vivo that could be used within animal models or clinical settings. Finally, we found that the NMS, MSN and eN3x9 TADs were well-expressed and potent in therapeutically important human cells.

While studies using CRISPR systems in combination with viral TADs have observed toxic effects at the cellular and organismal levels, it should be noted that our experiments do not conclusively demonstrate that viral TADs themselves are toxic. One remaining limitation that affects all synthetic gene activation platforms is that some loci may remain refractory to engineered transactivation, regardless of the effector deployed. This constraint likely stems from high basal expression levels at targeted sites[12,15] and/or other contextual factors that require further interrogation. Focused analyses at specific target sites, within specific cell types/organisms[80] and over longer time courses will likely be informative for optimized therapeutic proofs of concept and use cases.

In summary, we have used the rational redesign of natural human TADs to build synthetic transactivation modules that enable consistent and potent performance across programmable DNA binding platforms, mammalian cell types and genomic regulatory loci embedded within human chromatin. Although we used MTFs as sources of TADs here, our work establishes a framework that could be used with practically any natural or engineered TF and/or chromatin modifier in future efforts. The potency, small size, versatility and capacity for multiplexing of, and the lack of components from pathogenic human viruses associated with, the MSN, NMS and eN3x9 TADs and CRISPR-DREAM systems developed here could be valuable tools for fundamental and biomedical applications requiring potent and predictable activation of endogenous eukaryotic transcription.

## Online content

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

## Methods

### Cell culture
All experiments were performed within ten passages of cell stock thaws. HEK293T (CRL-11268), HeLa (CCL-2), A549 (CCL-185), SK-BR-3 (HTB-30), U2OS (HTB-96), HCT116 (CRL-247), K562 (CRL-243), CHO-K1 (CCL-61), ARPE-19 (CRL-2302), HFF (CRL-2429), Jurkat-T (TIB-152), hTERT-MSC (SCRC-4000) and Neuro-2a (CCL-131) cells were purchased from the American Type Culture Collection (ATCC) and cultured in ATCC-recommended media supplemented with 10% FBS (Sigma-Aldrich) and 1% pen/strep (100 U ml$^{-1}$ penicillin, 100 µg ml$^{-1}$ streptomycin; Gibco) at 37 °C and 5% $CO_2$. NIH3T3 cells were a kind gift from Dr. Caleb Bashor's laboratory and were cultured in DMEM supplemented with 10% FBS (Sigma-Aldrich) and 1% pen/strep (100 U ml$^{-1}$ penicillin, 100 µg ml$^{-1}$ streptomycin) at 37 °C and 5% $CO_2$.

### Plasmid transfection and nucleofection
HEK293T cell transfections were performed in 24-well plates using 375 ng of dCas9 expression plasmid and 125 ng of equimolar pooled or individual gRNAs/crRNAs. First, $1.25 \times 10^5$ HEK293T cells were plated the day before transfection and then transfected using Lipofectamine 3000 (Invitrogen) as per the manufacturer's instructions. For two-component systems (dCas9 + MCP or dCas9 + scFv systems), 187.5 ng of each plasmid was used. For multiplex gene activation experiments using DREAM platforms, 25 ng of each gRNA-encoding plasmid targeting each respective gene was used. Transfections in HeLa, A549, SK-BR-3, U2OS, HCT116, HFF, NIH3T3 and CHO-K1 were performed in 12-well plates using Lipofectamine 3000 and 375 ng of dCas9 plasmid, 375 ng of MCP-effector fusion proteins and 250 ng of DNA of MS2-modifed gRNA-encoding plasmid. For transfections using dCas12a fusion proteins where single genes were targeted, 375 ng of dCas12a-effector fusion plasmids and 125 ng of crRNA plasmids were transfected using Lipofectamine 3000 per the manufacturer's instructions. For multiplex gene activation experiments using dCas12a, 375 ng of dCas12a-effector fusion-encoding plasmid and 250 ng of multiplex crRNA expression plasmids were used. For experiments using *E. coli* and *Pseudomonas aeruginosa* Type I CRISPR systems, we followed the same stoichiometries used in previous studies[61,62]. For transfection of *ICAM1*-ZF effectors, 500 ng of each *ICAM1*-targeting ZF fusion was transfected. Transfections using *IL1RN*-TALE fusion proteins were performed using either 500 ng of a single TALE or a pool of four TALEs using 125 ng of each TALE fusion. All ZF and TALE transfections were performed in HEK293T cells in 24-well format using Lipofectamine 3000 as per the manufacturer's instructions. For K562 cells, $1 \times 10^6$ cells were nucleofected using the Lonza SF Cell Line 4D-Nucleofector Kit (Lonza V4XC-2012) and a Lonza 4D-Nucleofector (Lonza, AAF1002X) using the FF-120 program. In total, 2,000 ng of total plasmids were nucleofected in each condition using $1 \times 10^6$ K562 cells, and 667 ng each of dCas9 plasmid, MCP fusion plasmid and pooled MS2-sgRNA expression plasmid was nucleofected per condition. Immediately after nucleofection, K562 cells were transferred to prewarmed media-containing six-well plates. hTERT-MSCs were electroporated using the Neon transfection system (Thermo Fisher Scientific) using the 100-µl kit. In total, $5 \times 10^5$ hTERT-MSCs were resuspended in 100 µl of resuspension buffer R and 10 µg of total DNA (3.75 µg of dCas9, 3.75 µg of MCP-fusion effector plasmid and 2.5 µg of MS2-modifed gRNA-encoding plasmid). Electroporation was performed using the settings recommended by the manufacturer for MSCs: voltage: 990 V; pulse width: 40 ms; pulse number: 1. For fibroblast reprogramming experiments, we used the Neon transfection system using the amounts of endotoxin-free DNA described previously[17] and below. Dual AAV (500 ng of each) and AIO AAV (1 µg) construct transfections were performed in Neuro-2a cells in 12-well format using Lipofectamine 3000 as per the manufacturer's instructions.

### Peripheral blood mononuclear cell isolation, culture and nucleofection
De-identified white blood cell concentrates (buffy coats) were obtained from the Gulf Coast Regional Blood Center in Houston, Texas. Peripheral blood mononuclear cells (PBMCs) were isolated from buffy coats using Ficoll gradient separation and cryopreserved in liquid nitrogen until later use. Next, $1 \times 10^6$ PBMCs per well were stimulated for 48 h in a CD3/CD28 (Tonbo Biosciences, 700037U100 and 70289U100, respectively)-coated 24-well plate containing RPMI medium supplemented with 10% FBS (Sigma-Aldrich), 1% pen/strep (Gibco), 10 ng ml$^{-1}$ IL-15 (Tonbo Biosciences, 218157U002) and 10 ng ml$^{-1}$ IL-7 (Tonbo Biosciences, 218079U002). Stimulated PBMCs were electroporated using the Neon transfection system (Thermo Fisher Scientific) 100-µl kit per the manufacturer protocol. Briefly, PBMCs were centrifuged at 300$g$ for 5 min and resuspended in Neon Resuspension Buffer T to a final density of $1 \times 10^7$ cells per ml. Then, 100 µl of the resuspended cells ($1 \times 10^6$ cells) was mixed with 12 µg of total plasmid DNA (4.5 µg of dCas9 fusion-encoding plasmids, 4.5 µg of MCP fusion-encoding plasmids and 3 µg of four equimolar pooled MS2-modifed gRNA-encoding plasmids) and electroporated with the following program specifications using a 100-µl Neon Tip: pulse voltage: 2,150 V; pulse width: 20 ms; pulse number: 1. Endotoxin-free plasmids were used in all experiments. After electroporation, PBMCs were incubated in prewarmed six-well plates containing RPMI medium supplemented with 10% FBS (Sigma-Aldrich), 1% pen/strep (Gibco), 10 ng ml$^{-1}$ IL-15 and 10 ng ml$^{-1}$ IL-7. PBMCs were maintained at 37 °C, 5% $CO_2$ for 48 h before RNA isolation and quantitative reverse transcription PCR (qPCR).

### Human primary T cell and primary umbilical cord MSC culture and lentiviral transduction
PBMCs were isolated from de-identified white blood cell concentrates (buffy coats) using Ficoll gradient separation. T cells were isolated using negative selection via the EasySep Human T Cell Isolation Kit (STEMCELL, 17951). T cells were frozen in Bambanker Cell Freezing Media (Bulldog Bio, BB01) and stored in liquid nitrogen until use. Umbilical cord-derived MSCs (ATCC, PCS-500-010) were cultured in MSC basal medium (ATCC, PCS-500-030) supplemented with Mesenchymal Stem Cell Growth Kit (PCS-500-040) containing rhFGF basic (5 ng ml$^{-1}$), rhFGF acidic (5 ng ml$^{-1}$), rhEGF (5 ng ml$^{-1}$), FBS (2%) and L-alanyl-L-glutamine (2.4 mM). MSC medium was also supplemented with 1% pen/strep (Gibco, 15140122). MSCs were maintained at 37 °C, 5% $CO_2$. Lentiviral transduction was performed in stimulated T cells as previously described[81]. Briefly, $1 \times 10^6$ T cells per well were stimulated for 24 h with Dynabeads Human T-Activator CD3/CD28 for T Cell Expansion and Activation (Thermo Fisher Scientific, 11161D) according to the manufacturer's instructions in a 24-well plate containing X-VIVO 15 medium (Lonza, 04418Q) supplemented with 5% FBS (Sigma-Aldrich), 55 mM 2-mercaptoethanol (Gibco, 21985023), 4 mM N-acetyl-L-cysteine (Thermo Fisher Scientific, 160280250) and 500 IU ml$^{-1}$ recombinant human IL-2 (Biolegend, 589104). Stimulated T cells were co-transduced via spinoculation at 931$g$, 37 °C for 2 h in a plate coated with Retronectin (Takara Bio, T100B) with an MOI of ~5.0 for each lentivirus (dCas9 lentivirus at MOI ~5.0 and gRNA-MCP-fusion effector lentivirus). After spinoculation, T cells were maintained at 37 °C, 5% $CO_2$ for 48 h before downstream experiments. MSCs were co-transduced with an MOI of ~10.0 (dCas9 lentivirus at MOI ~10.0 and gRNA-MCP-fusion effector lentivirus at MOI ~10.0) for each lentivirus via reverse transduction, by seeding $1.25 \times 10^5$ cells into each well of a 12-well plate containing the virus in MSC medium supplemented with 8 µg ml$^{-1}$ polybrene. Medium was changed after 16 h. Further experimental analyses were performed at 72 h post-transduction.

### Mouse primary neuron culture and AAV8 transduction
Mouse C57 Cortex Neurons (Lonza, M-CX-300) were cultured in primary neuron basal medium supplemented with 2 mM L-glutamine,

GA-1000 and 2% NSF. In brief, $4 \times 10^5$ cells were seeded in poly-D-lysine- and laminin-coated 24-well plates and cultured for 7 d for neuronal differentiation. On day 8, cells from each well were transduced with $1 \times 10^{10}$ AAV8 viral particles ($2.5 \times 10^4$ per cell). At 5 d post-transduction, cells were collected for RNA isolation and qPCR analysis.

## Molecular cloning
Molecular cloning details are provided in Supplementary Note 6 and Supplementary Tables 1–6.

## Lentiviral packaging
All lentiviral transfer and packaging plasmids were purified using the Endofree Plasmid Maxi Kit (Qiagen, 12362). Lentivirus was packaged as previously described[81] with minor modifications. Briefly, HEK293T cells were seeded into 225-mm flasks and maintained in DMEM. OptiMem was used for transfection and sodium butyrate was added to a final concentration of 4 mM. Lentivirus was then concentrated 100× using the Lenti-X concentrator (Takara Bio, 631232). Biological titration of lentivirus by qPCR was carried out as previously described[82], with the following modifications: Volumes of 10, 5, 1, 0.1, 0.01 and 0 μl of concentrated lentiviral particles were reverse-transduced into $5 \times 10^4$ HEK293T cells with 8 μg ml$^{-1}$ polybrene (Millipore-Sigma, TR1003G) in 24-well format with medium exchanged after 14 h of transduction. gDNA was extracted at 96 h post-transduction using the DNeasy Blood & Tissue Kit (Qiagen, 69506). qPCR was performed using 67.5 ng of gDNA for each condition in 10-μl reactions using Luna Universal qPCR Master Mix (NEB, M3003E).

## Western blotting
Cells were lysed in RIPA buffer (Thermo Scientific, 89900) with 1 × protease inhibitor cocktail (Thermo Scientific, 78442), lysates were cleared by centrifugation and protein quantitation was performed using the BCA method (Pierce, 23225). Next, 15–30 μg of lysate was separated using precast 7.5% or 10% SDS–PAGE (Bio-Rad) and then transferred onto PVDF membranes using the Trans-Blot Turbo system (Bio-Rad). Membranes were blocked using 5% BSA in 1 × TBST and incubated overnight with primary antibody (anti-Cas9, 1:1,000 dilution, Diagenode no. C15200216; Anti-FLAG, 1:2,000 dilution, Sigma-Aldrich no. F1804; anti-β-Tubulin, 1:1,000 dilution, Bio-Rad no. 12004166). Then, membranes were washed with 1 × TBST three times (10 min each wash) and incubated with respective rabbit or mouse (CST no. 7074P2 or CST no. 7076S, respectively) HRP-tagged secondary antibodies (1:2,000 dilution) for 1 h. Next, membranes were washed with 1 × TBST three times (10 min each wash). Membranes were then incubated with ECL solution (Bio-Rad no. 1705061) and imaged using a Chemidoc-MP system (Bio-Rad). The β-tubulin antibody was tagged with Rhodamine (Bio-Rad no. 12004166) and was imaged using the Rhodamine channel in Chemidoc-MP as per the manufacturer's instructions. Uncropped blots are provided as Source data and in the Supplementary Information.

## Quantitative reverse-transcriptase PCR (qPCR)
RNA (including pre-miRNA) was isolated using the RNeasy Plus mini kit (Qiagen no. 74136). Then, 500–2,000 ng of RNA (quantified using a Nanodrop 3000C, Thermo Fisher) was used as a template for complementary DNA synthesis (Bio-Rad no. 1725038). cDNA was diluted 10× and 4.5 μl of diluted cDNA was used for each qPCR reaction in 10-μl reaction volumes. Real-time qPCR was performed using SYBR Green Master Mix (Bio-Rad no. 1725275) in the CFX96 Real-Time PCR system with a C1000 Thermal Cycler (Bio-Rad). Results are represented as fold change above control after normalization to *GAPDH* in all experiments using human cells. For murine cells, 18S ribosomal RNA was used for normalization. For CHO-K1 cells, *Gnb1* was used for normalization. Undetectable samples were assigned a cycle threshold ($C_t$) value of 45 cycles. All qPCR primers and cycling conditions are listed in Supplementary Table 7.

## Mature miRNA isolation and qPCR for miRNAs
Mature miRNA was isolated using the miRNA isolation kit (Qiagen no. 217084). Next, 500 ng of isolated miRNA was polyadenylated using polyA polymerase (Quantabio no. 95107) in 10-μl reactions per sample and then used for cDNA synthesis using qScript Reverse Transcriptase and oligo-dT primers attached to unique adapter sequences to allow specific amplification of mature miRNA using qPCR in a total 20-μl reaction (Quantabio no. 95107). cDNA was diluted and 10 ng of miRNA cDNA was used for qPCR in a 25-μl reaction volume. PerfeCTa SYBR Green SuperMix (Quantabio no. 95053), miR-146a-specific forward primer and PerfeCTa universal reverse primer were used to perform qPCR. U6 small nuclear RNA was used for normalization. All qPCR primers and cycling conditions are listed in Supplementary Table 7.

## Immunofluorescence microscopy
HFFs (CRL-2429, ATCC) and HFF-derived iPSCs were grown in Gel-trex (Gibco, A1413302)-coated 12-well plates and were fixed with 3.7% formaldehyde and then blocked with 3% BSA in 1 × PBS for 1 h at room temperature before imaging. Primary antibodies for SSEA4 (CST no. 43782), TRA-1-60 (CST no. 61220) and TRA-1-81 (CST no. 83321) were diluted (1:200) in 1% BSA in 1 × PBS and incubated overnight at 4 °C. The next day, cells were washed with 1 × PBS, incubated with appropriate Alexaflour-488-conjugated secondary antibodies (CST no. A21042 and CST no. A21151) (1:500 dilution) for 1 h at room temperature and then washed again three times with 1 × PBS. Cells were then incubated with DAPI (Invitrogen no. D1306)-containing PBS (100 nM final concentration) for 10 min, washed three times with 1 × PBS and then imaged using a Nikon ECLIPSE Ti2 fluorescence microscope.

## CD34 surface expression analysis
Surface staining of CD34 in HEK293T cells was performed using CD34-PE antibody (Invitrogen, no. MA1-10205). In brief, at 72 h post-transfection, cells from 24-well plates were detached using TrypLE Select (Gibco, no. 12563011). Single-cell suspensions were washed with complete media and then with 0.22 μM filtered 1 × FACS buffer (1% BSA in 1 × PBS). Next, cells were incubated with CD34-PE antibody (20 μl per $10^6$ cells) or IgG-PE isotype antibody (Invitrogen, no. 12-4714-42) in 1 × FACS buffer for 30 min. Stained cell fluorescence intensity was measured using a Sony SA3800 spectral analyzer. To assess the CD34 expression in EGFP-positive MCP-effector transfected cells, single cells were gated based on EGFP expression and assessed for CD34 expression. Data were analyzed using FlowJo software (v.10). First, cell debris and doublets were excluded with the FSC-A/SSC-A dot plot, followed by the FSC-H/FSC-A dot plot. Events were excluded with intensity $<1 \times 10^{-2}$ using FSC-A/PE-A dot plots. The CD34 surface marker-positive cells were gated using unstained or isotype controls. For cells transfected with an EGFP-expressing effector construct, a similar gating strategy was used to select single cells as above. Events were excluded with intensity $<1 \times 10^{-2}$ using PE-A/EGFP-A dot plots, then EGFP-expressing cells were analyzed for CD34 surface marker expression using unstained or isotype controls.

## CUT&RUN
CUT&RUN was performed using the Epicypher CUTANA ChIC/CUT&RUN Kit (Epicypher, no. 14-1048). Briefly, transfected cells were detached and collected using TrypLE Select (Gibco, no. 12563011), washed once with 1 × PBS and then dissolved in 300 μl of wash buffer. Next, each of three 100-μl aliquots (~1/3 of each 24-well plate) of cells were processed for H3K4me3 antibody (Epicypher, no. 13-0041), H3K27ac antibody (Epicypher, no. 13-0045) or input DNA, respectively. Cells were first immobilized on concanavalin A beads, and then incubated with respective antibody (0.5 μg per sample) overnight at 4 °C in antibody dilution buffer (cell permeabilization buffer + EDTA). On the following day, cells were washed twice with cell permeabilization buffer. After washing the beads, pAG-MNase was added to

the immobilized cells and then incubated for 2 h at 4 °C to digest and release DNA. For CUT&RUN–qPCR assays, purified DNA from both H3K4me3 antibody- and H3K27ac antibody-incubated samples was then assayed using qPCR. Relative enrichment of H3K4me3 and H3K27ac was expressed as fold change above control cells transfected with dCas9 + MCP-mCherry plasmid and after normalization to purified input DNA. qPCR primers used for CUT&RUN are shown in Supplementary Table 8.

### Generation of mini-DREAM component-expressing HEK293T cell line

HEK293T cells were co-transduced with HNH-deleted dCas9 and MCP-eN3x9-T2A-EGFP lentiviruses (each with an MOI of ~5.0) using 8 μg ml⁻¹ polybrene (Millipore-Sigma, TR1003G) in 24-well format. Medium was exchanged at 14 h post-transduction. Mini-DREAM HEK293T cells were then transfected with gRNA/gRNA array for further experimentation.

### Progesterone ELISA

Secreted progesterone was measured using the Progesterone Competitive ELISA Kit (Invitrogen. no. EIAP4C21). In brief, at 72 h post-transfection of control gRNA or the indicated MS2-gRNA array into a mini-DREAM-expressing HEK293T cell line, 50 μl of cell culture supernatant was directly used for ELISA as per the manufacturer's instructions, along with all recommended progesterone standards. Standard curves were generated using the polynomial function and progesterone concentration was determined and expressed in ng ml⁻¹.

### Fibroblast reprogramming

HFFs were cultured in 1 × DMEM supplemented with 1 × Glutamax (Gibco, 35050061) for two passages before nucleofection with respective components. Cells were grown in 15-cm dishes (Corning) and detached using TrypLE Select (Gibco, no. 12563011). Single-cell suspensions were washed with complete media and then with 1 × PBS. For each 1 × 10⁶ cells, a total of 6 μg of endotoxin-free plasmids (Macherey-Nagel, 740424; 2 μg of CRISPR activator plasmid, 2 μg of pluripotency factor targeting gRNA plasmid and 2 μg of EEA-motif targeting gRNA expression plasmids) was nucleofected using a 100-μl Neon transfection tip in R buffer using the following settings: 1,650 V, 10 ms and 3 pulses. Nucleofected fibroblasts were then immediately transferred to Geltrex (Gibco)-coated 10-cm cell culture dishes in prewarmed medium. The next day, medium was exchanged. After 4 d, medium was replaced with iPSC induction medium[17]. Induction medium was then exchanged every other day for 18 d. After 18 d, iPSC colonies were counted, and colonies were picked using sterile forceps and then transferred to Geltrex-coated 12-well plates. iPSC colonies were maintained in complete E8 medium and passaged as necessary using ReLeSR passaging reagent (STEMCELL, no. 05872). RNA was isolated from iPSC clones using the RNeasy Plus Mini Kit (Qiagen, no. 74136) and colonies were immunostained using indicated antibodies and counterstained with DAPI (Invitrogen) for nuclear visualization.

### RNA-seq

RNA-seq was performed in duplicate for each experimental condition. At 72 h post-transfection, RNA was isolated using the RNeasy Plus Mini Kit (Qiagen). RNA integrity was first assessed using a Bioanalyzer 2200 (Agilent) and then RNA-seq libraries were constructed using the TruSeq Stranded Total RNA Gold (Illumina, RS-122-2303). The qualities of RNA-seq libraries were verified using the Tape Station D1000 assay (Tape Station 2200, Agilent Technologies) and the concentrations of RNA-seq libraries were checked again using real-time PCR (QuantStudio 6 Flex Real-Time PCR System, Applied Biosystem). Libraries were normalized and pooled before sequencing. Sequencing was performed using an Illumina Hiseq 3000 with paired-end 75-base-pair reads. Reads

were aligned to the human genome (hg38) Gencode Release 36 reference using STAR aligner (v.2.7.3a). Transcript levels were quantified to the reference genome using a Bayesian approach. Normalization was done using the counts per million method. Differential expression was done using DESeq2 (v.3.5) with default parameters. Genes were considered significantly differentially expressed based upon a fold change >2 or <−2 and a false discovery rate < 0.05.

### 9aa TAD prediction

9aa TADs were predicted using previously described software (https://www.med.muni.cz/9aaTAD/)[55] using the 'moderately stringent pattern' criteria and all 'refinement criteria', and only TADs with 100% matches were then selected for evaluation in MCP fusion proteins.

### Cytotoxicity assays

Cellular toxicity assays in primary T cells were performed at 72 h post-transduction using the Annexin V:PE Apoptosis Detection Kit (BD Biosciences, 559763). In brief, cells were stained with 7-AAD and Annexin V:PE according to the manufacturer's protocol. Stained cell fluorescence was measured using a Sony SA3800 spectral analyzer. EGFP-positive single cells were gated and assessed for 7-AAD and Annexin V:PE fluorescence. All conditions were measured in biological triplicate and in technical duplicate. The toxicity of treatment groups was compared with the negative control (dCas9 alone), and camptothecin (5 mM) and 65 °C heat shock were used as positive controls of apoptosis and membrane permeability, respectively. Cellular toxicity assays in HEK293T and U2OS cells were performed using Hoechst and 7-AAD staining followed by microscopy. In brief, at 48 h post-transfection of different CRISPRa tools, medium was aspirated from each 24-well plate and 150 μl of staining solution was gently added to cover the cells in each condition. The staining solution contained Hoechst 33342 (Thermo Scientific, no. 62249) diluted 1:6,000 and 5 μl of 7-AAD (BD Biosciences, no. 51-68981E) in sterile 1 × PBS. The 24-well plates were then incubated for 30 min at room temperature while protected from light. After incubation, automated images were taken of cells using a Nikon Ti2-E inverted microscope equipped with an Andor Zyla 4.2 sCMOS camera and 488-nm and 561-nm lasers using the 10X PLAN APO λD objective.

### Statistics and reproducibility

No statistical methods were used to predetermine the sample sizes. No data were excluded from analyses. Randomization is not relevant to this study. The investigators were not blinded to allocation during experiments and outcome assessment. Individual data points are represented as dots in bar graphs. The western blot image shown in Fig. 1c is representative of two independent biological experiments. The microscopic image shown in Fig. 5a is representative of three independent biological experiments. The microscopic images shown in Fig. 5d,e and Supplementary Fig. 31j are representative of two independent biological experiments. The microscopic images shown in Fig. 6a,b are representative of three independent biological experiments. The microscopic image shown in Supplementary Fig. 35d is representative of four independent biological replicates. The microscopic images shown in Supplementary Fig. 34a,c are representative of three independent biological replicates. All data used for statistical analysis relied upon a minimum of three independent biological replicates. Error bars indicate mean ± s.e.m. Statistical analyses of qPCR data were conducted using Student's t-tests as indicated in figure legends. Results were considered statistically significant when the P value was <0.05. All bar graphs, error bars and statistics were generated using GraphPad Prism v.9.0.

### Reporting summary

Further information on research design is available in the Nature Portfolio Reporting Summary linked to this article.

## Data availability

Plasmids encoding MCP-MSN, MCP-NMS, MCP-eN3x9, MCP-3x 9aa TAD, MS2 stem loop modified gRNA backbone for CjCas9 and AAV vector components are available via Addgene. The RNA-sequencing data used in this study have been deposited to the Gene Expression Omnibus under accession number GSE238178. The human reference genome GRCh38 and the mouse reference genome mm10 are publicly available. Source data, including data used for statistical analyses for all figures and supplementary figures, are provided with this paper.

## Code availability

No custom code was used in this study. All analyses were performed using standard workflows and open-source software as described in the Methods section.

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

## Acknowledgements

We thank J. de Rossi and D. Jain for their assistance with 9aa TADs prediction and cloning. We thank A. Sarkar for her help with NGS data analysis and the selection of miRNA target loci in HEK293T cells. We thank H. Deshmukh and S. Mishra for their assistance with flow cytometry and protein sequence analyses, respectively. We also thank A. Pickar-Oliver and C. A. Gersbach for providing *E. coli* Type I CRISPR Cascade plasmids. We thank all members of the Hilton lab for discussions and insights. This work was supported by a Cancer Prevention & Research Institute of Texas (CPRIT) Award (grant no. RR170030) and NIH Awards (grant nos. R35GM143532, R21EB030772 and R56HG012206) to I.B.H. M.E. was supported by the American Heart Association predoctoral fellowship program. R.S.G.-R. was supported by the Fulbright program and the National Council of Science and Technology of Mexico. H.S. was supported by the NSF GRFP.

## Author contributions

B.M. and I.B.H. conceived the project and designed experiments. B.M. performed most experiments and analyzed the data with the assistance of A.C., D.A.B., R.S.G.-R., J.L., J.G., K.W., Y.G., M.E., A.K.P., H.S., G.B., D.R.R. and S.K.. B.M. and I.B.H. wrote the manuscript with input from all authors.

## Competing interests

B.M., J.G. and I.B.H. have filed a patent related to this work. I.B.H. has filed patent applications related to other CRISPR technologies for genome engineering. The remaining authors declare no competing interests.

## Additional information

**Correspondence and requests for materials** should be addressed to Isaac B. Hilton.

# Reporting Summary

## Statistics

For all statistical analyses, confirm that the following items are present in the figure legend, table legend, main text, or Methods section.

| n/a | Confirmed | |
|---|---|---|
| ☐ | ☒ | The exact sample size (*n*) for each experimental group/condition, given as a discrete number and unit of measurement |
| ☐ | ☒ | A statement on whether measurements were taken from distinct samples or whether the same sample was measured repeatedly |
| ☐ | ☒ | The statistical test(s) used AND whether they are one- or two-sided *Only common tests should be described solely by name; describe more complex techniques in the Methods section.* |
| ☒ | ☐ | A description of all covariates tested |
| ☒ | ☐ | A description of any assumptions or corrections, such as tests of normality and adjustment for multiple comparisons |
| ☐ | ☒ | A full description of the statistical parameters including central tendency (e.g. means) or other basic estimates (e.g. regression coefficient) AND variation (e.g. standard deviation) or associated estimates of uncertainty (e.g. confidence intervals) |
| ☐ | ☒ | For null hypothesis testing, the test statistic (e.g. *F*, *t*, *r*) with confidence intervals, effect sizes, degrees of freedom and *P* value noted *Give P values as exact values whenever suitable.* |
| ☐ | ☒ | For Bayesian analysis, information on the choice of priors and Markov chain Monte Carlo settings |
| ☒ | ☐ | For hierarchical and complex designs, identification of the appropriate level for tests and full reporting of outcomes |
| ☒ | ☐ | Estimates of effect sizes (e.g. Cohen's *d*, Pearson's *r*), indicating how they were calculated |

*Our web collection on statistics for biologists contains articles on many of the points above.*

## Software and code

Policy information about availability of computer code

Data collection   All spacer sequences for SpCas9 systems were designed using the Custom Alt-R® CRISPR-Cas9 guide RNA design tool (IDT).
Illumina Hiseq3000 was used for high throughput RNA sequencing.
Nikon ECLIPSE Ti2 fluorescent microscope was used for collecting microscopic (both Phase contrast and fluorescent) images.
BioRad Chemidoc-MP was used for collecting Western Blot data.
BioRad CFX96 Real-Time PCR system with a C1000 Thermal Cycler was used for QPCR data collection.
9aa TADs were predicted using previously described online tool http://www.at.embnet.org/toolbox/9aatad/.
Sony SA3800 spectral analyzer was used for collecting CD34 surface staining data.

| Data analysis | All data analysis, presented graph and statistics were performed using GraphPad Prism 9.0 software except data presented on Fig. 1c, Fig. 1f, Fig. 5a, Fig. 5d, Fig. 5e, Fig. 6a, Fig. 6b and Supplementary Fig. 11b, Supplementary Fig. 10a, Supplementary Fig. 11d, Supplementary Fig. 10e,Supplementary Fig. 12b, Supplementary Fig. 12d, Supplementary Fig. 26b, Supplementary Fig. 31j, Supplementary Fig. 33a, Supplementary Fig. 33b, Supplementary Fig. 33c, Supplementary Fig. 34a, Supplementary Fig. 34c, Supplementary Fig. 35d. RNA Sequencing was performed using an Illumina Hiseq 3000 with paired end 75 base pair reads. Reads were aligned to the human genome (hg38) Gencode Release 36 reference using STAR aligner (v2.7.3a). Transcript levels were quantified to the reference genome using a Bayesian approach. Normalization was done using counts per million (CPM) method. Differential expression was done using DESeq2 (v3.5) with default parameters. Analysis of DNA sequences and construction of virtual plasmid maps were performed using SnapGene. For QPCR analysis CFX Manager software (Bio-Rad CFX Maestro 2.2) was used. Flow Jo v10.8.1 was used for flow cytometry data analysis. All data arrangement for preparation of Figures were performed on Adobe Illustrator 2021. |
|---|---|

For manuscripts utilizing custom algorithms or software that are central to the research but not yet described in published literature, software must be made available to editors and reviewers. We strongly encourage code deposition in a community repository (e.g. GitHub). See the Nature Portfolio guidelines for submitting code & software for further information.

## Data

Policy information about availability of data

All manuscripts must include a data availability statement. This statement should provide the following information, where applicable:
- Accession codes, unique identifiers, or web links for publicly available datasets
- A description of any restrictions on data availability
- For clinical datasets or third party data, please ensure that the statement adheres to our policy

Plasmids encoding MCP-MSN, MCP-NMS, MCP-eN3x9, MCP-3x 9aa TAD, MS2 stem loop modified gRNA backbone for CjCas9, and AAV vector components are available via Addgene. The RNA-sequencing data used in this study have been deposited to Gene Expression Omnibus under the accession number GSE238178. The Human reference genome GRCh38 and the mouse reference genome GRCm38 are publicly available. Source data including data used for statistical analyses are provided for all Figures and Supplementary Figures. All other datasets generated or analyzed are available upon reasonable request.

## Human research participants

Policy information about studies involving human research participants and Sex and Gender in Research.

| Reporting on sex and gender | N/A |
|---|---|
| Population characteristics | N/A |
| Recruitment | N/A |
| Ethics oversight | N/A |

Note that full information on the approval of the study protocol must also be provided in the manuscript.

# Field-specific reporting

Please select the one below that is the best fit for your research. If you are not sure, read the appropriate sections before making your selection.

☒ Life sciences        ☐ Behavioural & social sciences        ☐ Ecological, evolutionary & environmental sciences

For a reference copy of the document with all sections, see nature.com/documents/nr-reporting-summary-flat.pdf

# Life sciences study design

All studies must disclose on these points even when the disclosure is negative.

| Sample size | No statistical methods were applied to predetermine sample size. Sample sizes are similar to previous relevant publications. |
|---|---|
| Data exclusions | No data was excluded. No method was applied for data exclusion. |
| Replication | All experiment contains at least two biological replicates. All attempts were succesful. |
| Randomization | Experiments were neither randomized or blinded. Randomization is not relevant to this study. |
| Blinding | Data collection and analysis were not performed blind to the conditions of the experiments and outcome assessment. |

# Reporting for specific materials, systems and methods

We require information from authors about some types of materials, experimental systems and methods used in many studies. Here, indicate whether each material, system or method listed is relevant to your study. If you are not sure if a list item applies to your research, read the appropriate section before selecting a response.

## Materials & experimental systems

| n/a | Involved in the study |
|---|---|
| ☐ | ☒ Antibodies |
| ☐ | ☒ Eukaryotic cell lines |
| ☒ | ☐ Palaeontology and archaeology |
| ☒ | ☐ Animals and other organisms |
| ☒ | ☐ Clinical data |
| ☒ | ☐ Dual use research of concern |

## Methods

| n/a | Involved in the study |
|---|---|
| ☒ | ☐ ChIP-seq |
| ☐ | ☒ Flow cytometry |
| ☒ | ☐ MRI-based neuroimaging |

## Antibodies

| | |
|---|---|
| Antibodies used | anti-Cas9; Diagenode #C15200216, Lot. No.#4 (1:1000 dilution), Anti-FLAG; Sigma-Aldrich #F1804, Lot. No.#SLCM4081 (1:2000 dilution), anti-β-Tubulin; Bio-Rad #12004166, Batch No.64512248, (1:1000 dilution) were used as primary antibodies for Western Blot. Anti-rabbit IgG, HRP-linked secondary Antibody; Cell Signaling Technology #7074P2, Lot. No.#28, (1:2000 dilution), Anti-mouse IgG, HRP-linked secondary Antibody; Cell Signaling Technology #7076S, Lot. No.#33 (1:2000 dilution) were were used as secondary antibodies for Western Blot.<br>SSEA-4, CST #43782, Lot. No.#2, (1:200 dilution), TRA1-60, CST #61220, Lot. No.#1, (1:200 dilution) and TRA1-81, Lot. No.#1, (1:200 dilution), CST #83321 were used for immunofluorescence microscopy. Alexa Flour 488 goat anti-mouse IgM ; Invitrogen, #A21042, Lot No.# 2306815 (1:500 dilution) and Alexa Flour 488 goat anti-mouse IgG3 ; Invitrogen, #A21151, Lot No.# 2311808 (1:500 dilution) were used as secondary antibodies for immunofluorescence microscopy.<br>CD34-PE; Invitrogen, #MA1-10205, Clone # QBEND/10, Lot. No.#YA3803041A was used (20 µL/100,000 cells) for surface staining of CD34 in HEK293T cells.<br>H3K4me3 antibody; Epicypher, #13-0041, Lot. No.#21337004-12 (0.5 µg/condition), H3K27ac antibody ;Epicypher, #13-0045, Lot. No.#22242006-81 (0.5 µg/condition) were used for CUT&RUN assay. |
| Validation | Cas9 : https://www.diagenode.com/en/p/crispr-cas9-monoclonal-antibody-4g10-100-ug. Also validated in "Systematic comparison of CRISPR-based transcriptional activators uncovers gene-regulatory features of enhancer-promoter interactions. Nucleic Acids Res (2022) by Wang, K. et al."<br>FLAG : https://www.sigmaaldrich.com/US/en/product/sigma/f1804. Also validated in "Systematic comparison of CRISPR-based transcriptional activators uncovers gene-regulatory features of enhancer-promoter interactions. Nucleic Acids Res (2022) by Wang, K. et al."<br>β-Tubulin : https://www.bio-rad.com/en-us/sku/12004166-hfab-rhodamine-anti-tubulin-primary-antibody-40-ul?ID=12004166. Also validated in "Systematic comparison of CRISPR-based transcriptional activators uncovers gene-regulatory features of enhancer-promoter interactions. Nucleic Acids Res (2022) by Wang, K. et al."<br>Anti-rabbit IgG, HRP-linked secondary Antibody : https://www.cellsignal.com/products/secondary-antibodies/anti-rabbit-igg-hrp-linked-antibody/707. Also validated in "Systematic comparison of CRISPR-based transcriptional activators uncovers gene-regulatory features of enhancer-promoter interactions. Nucleic Acids Res (2022) by Wang, K. et al."<br>Anti-mouse IgG, HRP-linked secondary Antibody: https://www.cellsignal.com/products/secondary-antibodies/anti-mouse-igg-hrp-linked-antibody/7076. Also validated in "Systematic comparison of CRISPR-based transcriptional activators uncovers gene-regulatory features of enhancer-promoter interactions. Nucleic Acids Res (2022) by Wang, K. et al."<br>SSEA4, TRA1-60 and TRA1-81 : https://media.cellsignal.com/pdf/9656.pdf. Further validated in this study- Liberski, A.R. et al. (2013) J Proteome Res 12, 3233-45.<br>Alexa Flour 488 goat anti-mouse IgM : https://www.thermofisher.com/antibody/product/Goat-anti-Mouse-IgM-Heavy-chain-Cross-Adsorbed-Secondary-Antibody-Polyclonal/A-21042<br>Alexa Flour 488 goat anti-mouse IgG3 : https://www.thermofisher.com/antibody/product/Goat-anti-Mouse-IgG3-Cross-Adsorbed-Secondary-Antibody-Polyclonal/A-21151<br>CD34: https://www.thermofisher.com/order/genome-database/dataSheetPdf?producttype=antibody&productsubtype=antibody_primary&productId=MA1-10205&version=300<br>H3K4me3 antibody (Epicypher, #13-0041): https://www.epicypher.com/products/epigenetics-reagents-and-assays/cutana-chic-cut-and-run-kit. This antibody was included as a positive control in the Epicypher CUTANA ChIC/CUT&RUN Kit (Epicypher, #14-1048).<br>H3K27ac antibody (Epicypher, #13-0045): https://www.epicypher.com/products/antibodies/snap-chip-certified-antibodies/histone-H3K27ac-antibody-snap-chip-certified. Also validated in "Systematic comparison of CRISPR-based transcriptional activators uncovers gene-regulatory features of enhancer-promoter interactions. Nucleic Acids Res (2022) by Wang, K. et al." |

## Eukaryotic cell lines

Policy information about cell lines and Sex and Gender in Research

| | |
|---|---|
| Cell line source(s) | HEK293T (ATCC, CRL-11268), HeLa (ATCC, CCL-2), A549 (ATCC, CCL-185), SK-BR-3 (ATCC, HTB-30), U2OS (ATCC, HTB-96), HCT116 (ATCC, CRL-247), K562 (ATCC, CRL-243), CHO-K1 (ATCC, CCL-61), ARPE-19 (ATCC, CRL-2302), HFF (ATCC, CRL-2429), |

Jurkat-T (ATCC, TIB-152), Neuro-2A (ATCC, CCL-131), and hTERT-MSC (ATCC, SCRC-4000) cells were purchased from American Type Cell Culture (ATCC, USA). NIH3T3 cells were a kind gift from Dr. Caleb Bashor's lab and similar to ATCC, CRL-1658.

Authentication
Cell lines from ATCC were authenticated by STR analysis.

Mycoplasma contamination
No Mycoplasma contamination was observed.

Commonly misidentified lines
(See ICLAC register)
None.

# Flow Cytometry

## Plots

Confirm that:

☒ The axis labels state the marker and fluorochrome used (e.g. CD4-FITC).

☒ The axis scales are clearly visible. Include numbers along axes only for bottom left plot of group (a 'group' is an analysis of identical markers).

☒ All plots are contour plots with outliers or pseudocolor plots.

☒ A numerical value for number of cells or percentage (with statistics) is provided.

## Methodology

Sample preparation
Up to 100,000 Human Primary T cells were analyzed. All samples from respective groups were prepared similarly. 72 h post-transduced human primary T cells were processed using the Annexin V: PE Apoptosis Detection Kit according to manufacturer's instructions. For CD34 surface staining single HEK293T Single cell suspensions were washed with complete media and then with 0.22 um filtered 1X FACS Buffer (1% BSA in 1X PBS). Next, cells were incubated with CD34-PE antibody (20 µL/106 cells) or IgG-PE isotype antibody in 1X FACS Buffer for 30 mins. Stained cell fluorescence intensity was measured using a Sony SA3800 spectral analyzer.

Instrument
Sony SA3800 Spectral Snalyzer and Flow Cytometer

Software
Flow Jo v10.8.1

Cell population abundance
Live cell represents >75% of total collected event . EGFP-positive events represent between 1% and 50% of the total human primary T cell population.

Gating strategy
To assess the CD34 expression in EGFP positive MCP-effector transfected cells, single cells were gated based on EGFP expression. First cell debris and doublets were excluded with the FSC-A/SSC-A dot plot, followed by FSC-H/FSC-A dot plot. Events were excluded with intensity <1X 10-2, using FSC-A/PE-A dot plots. The CD34 surface marker-positive cells were gated using unstained or isotype controls. For cells transfected with an EGFP expressing effector construct, similar gating strategy was used to select singles cells as above and then events were excluded with intensity <1X 10-2, using PE-A/EGFP-A dot plots, then EGFP expressing cells were analyzed for CD34 surface marker expression using unstained or isotype controls.For primary T cells, events were plotted by FSC-A and SSC-A, and the gate "Live cells" was drawn to include live cells and exclude potential debris/dead cells. From the "Live cells" population, events were plotted by FSC-H and FSC-A to gate for population "Single Cells". From the "Single Cells" population, events were plotted to show EGFP-positive signal and were subsequently gated in the "EGFP-positive cells" population. This population was then gated for PE:Annexin V signal in the x-axis and 7-AAD signal in the y-axis, to gate for four different cell populations: Early apoptosis/ Membrane integrity is present (PE+ , 7-AAD-), Late apoptosis/Already dead (PE+ , 7-AAD+), Cells with damaged membrane (PE- , 7-AAD+), and Viable cells/ No measurable apoptosis (PE- , 7-AAD-). Gates used are shown in Supplementary Figure 33 and source data for Supplementary Figure 33.

☒ Tick this box to confirm that a figure exemplifying the gating strategy is provided in the Supplementary Information.

