## [Peer Review File · Nature Methods]

Peer Review Information

Manuscript Title: Compact engineered human mechanosensitive transactivation modules enable potent and versatile synthetic transcriptional control

Corresponding author name(s): Isaac Hilton

Reviewer Comments & Decisions:

Decision Letter, initial version:
--

Dear Isaac,

Your Article, "Compact engineered human mechanosensitive transactivation modules enable potent and versatile synthetic transcriptional control", has now been seen by 3 reviewers. As you will see from their comments below, although the reviewers find your work of considerable potential interest, they have raised a number of concerns. We are interested in the possibility of publishing your paper in Nature Methods, but would like to consider your response to these concerns before we reach a final decision on publication.

We therefore invite you to revise your manuscript to address these concerns. In particular, please address the utility and versatility of this method over existing methods.

* include a point-by-point response to the reviewers and to any editorial suggestions

- * please underline/highlight any additions to the text or areas with other significant changes to facilitate review of the revised manuscript
- * address the points listed described below to conform to our open science requirements
- * ensure it complies with our general format requirements as set out in our guide to authors at www.nature.com/naturemethods
- * resubmit all the necessary files electronically by using the link below to access your home page

[REDACTED]

We hope to receive your revised paper within eight weeks. If you cannot send it within this time, please let us know. In this event, we will still be happy to reconsider your paper at a later date so long as nothing similar has been accepted for publication at Nature Methods or published elsewhere.

OPEN SCIENCE REQUIREMENTS

REPORTING SUMMARY AND EDITORIAL POLICY CHECKLISTS

Please note that these forms are dynamic ‘smart pdfs’ and must therefore be downloaded and completed in Adobe Reader. We will then flatten them for ease of use by the reviewers. If you would like to reference the guidance text as you complete the template, please access these flattened versions at <http://www.nature.com/authors/policies/availability.html>.

DATA AVAILABILITY

All novel DNA and RNA sequencing data, protein sequences, genetic polymorphisms, linked genotype and phenotype data, gene expression data, macromolecular structures, and proteomics data must be deposited in a publicly accessible database, and accession codes and associated hyperlinks must be provided in the “Data Availability” section.

Please include a “Data availability” subsection in the Online Methods. This section should inform readers about the availability of the data used to support the conclusions of your study, including accession codes to public repositories, references to source data that may be published alongside the paper, unique identifiers such as URLs to data repository entries, or data set DOIs, and any other statement about data availability. At a minimum, you should include the following statement: “The data that support the findings of this study are available from the corresponding author upon request”, describing which data is available upon request and mentioning any restrictions on availability. If DOIs are provided, please include these in the Reference list (authors, title, publisher (repository name), identifier, year). For more guidance on how to write this section please see:

<http://www.nature.com/authors/policies/data/data-availability-statements-data-citations.pdf>

CODE AVAILABILITY

Please include a “Code Availability” subsection in the Online Methods which details how your custom code is made available. Only in rare cases (where code is not central to the main conclusions of the paper) is the statement “available upon request” allowed (and reasons should be specified).

MATERIALS AVAILABILITY

SUPPLEMENTARY PROTOCOL

To help facilitate reproducibility and uptake of your method, we ask you to prepare a step-by-step Supplementary Protocol for the method described in this paper. We [encourage authors to share their step-by-step experimental protocols](https://www.nature.com/nature-research/editorial-policies/reporting-standards#protocols) on a protocol sharing platform of their choice and report the protocol DOI in the reference list. Nature Portfolio's Protocol Exchange is a free-to-use and open resource for protocols; protocols deposited in Protocol Exchange are citable and can be linked from the published article. More details can found at www.nature.com/protocolexchange/about.

ORCID

Nature Methods is committed to improving transparency in authorship. As part of our efforts in this

direction, we are now requesting that all authors identified as 'corresponding author' on published papers create and link their Open Researcher and Contributor Identifier (ORCID) with their account on the Manuscript Tracking System (MTS), prior to acceptance. This applies to primary research papers only. ORCID helps the scientific community achieve unambiguous attribution of all scholarly contributions. You can create and link your ORCID from the home page of the MTS by clicking on 'Modify my Springer Nature account'. For more information please visit www.springernature.com/orcid.

Sincerely,
Madhura

Madhura Mukhopadhyay, PhD
Senior Editor
Nature Methods

Reviewers' Comments:

Reviewer #1:

Remarks to the Author:

This work outlines the the design and validation of a multipartite activation domain derived from human mechanosensitive transcription factors (MTFs) to be used in conjunction with CRISPR-dCas9. Validation of constructs involved a panel of cells and genetic targets (including noncoding genomic regulatory elements) as well as a multitude of DNA binding (zinc finger and TALENs) and CRISPR based (Type I, II, and V) platforms. Applications including iPSC fibroblast reprogramming and both dual and all-in-one (AIO) AAV packaging were demonstrated as proof of concept.

To the best of my knowledge, there have been no prior works investigating MTF TADs in the context of CRISPRa to this extent. In this work, comparisons against state of the art CRISPRa platforms were used throughout as effective positive controls in a variety of experiments. Final tripartite MSN, NMS, and eN3x9 modules saw relatively comparable and moderate increases in the majority of in vitro experiments performed, lending credibility to both novelty and feasibility as an alternative to current

methods. Unique strengths of this method lie notably with increased cell viability in primary T cells and considerably improved performance with the SunTag and dCas12a platforms. Beneficial investigations included the evaluation of a compact construct involving minimized domains of the MTF TADs used in the full-size DREAM construct (similar to AIO miniCAFE development), methods for AIO delivery, and equivalent efficiency with readily modular integration of the multipartite domain to various existing modalities used for gene activation (ZFs and TALENs).

As of current implementation, this multipartite domain seemingly only holds true novel utility above existing platforms in the niche areas of aforementioned unique strengths. In particular, VP64, VP192, VPR, and p65-HSF1 (SAM) platforms still maintain comparable effectiveness (or in the case of VP192, seemingly improved performance in iPSC colony growth) in the majority of use cases explored and would likely see no inherent downsides in in vitro study. A specific benefit could be the easier integration into studies aiming for primarily human-derived platforms (like zinc fingers discussed in this study) or those concerned with immunogenicity/cytotoxicity in general. The versatility of this domain to replace many of the state-of-the-art platforms without significant losses (unlike p65-HSF1 which appears to lose efficiency when fused) combined with the outlined framework for validation and minimization, facilitate utility and the prospects of a novel technology respectively, but begs an inquiry into how existing platforms would fair if subjected to the same evaluations and combinatorics. As these evaluations do not currently exist, there is, by absence, novelty of the actual multipartite domain developed.

This work provides a thorough analytical framework and progression to both development and validation of the tripartite domains, suggesting reasonable utility as an equivalent alternative to existing CRISPRa platforms. However, it is noted that the strong benefits of the framework (ie. facilitating overcoming notable challenges in this technology), T-cell viability, AIO AAV delivery, and a composition devoid of viral components derive their importance from the strong benefits implied for in vivo work, without explicit in vivo investigation here.

Overall, all claims and statements appear reasonable and are contained within the results shown; and there is novelty in the domains, but activity over and above existing domains is not observed, so broader utility (beyond niche applications around managing immunogenicity) remains uncertain.

Minor Revisions and Recommendations:

Benchmark data vs MCP-p65-HSF instead of SAM could be desirable. Supplementary Figure 7 seems to demonstrate that MSN compared to p65-HSF1 without the viral VP64 fusion component of the SAM system is comparable to MSN and NMS. Potency of p65-HSF1 as an activation domain with recruitment may prove comparable to the tripartite fusion and would remain non-viral. Furthermore, potential changes to fusion conformations may see improvements similar to those seen with NMS and MNS.

An overall suggestion is to indicate the fold change differences like in Supplementary Fig. 11d to better emphasize significant improvements where apparent across results and claims.

Typo, Main-Page 5: sacas9 to sadcas9

Typo, Supplementary Figure 21a: ca9 to cas9

Figure 3. Suggest a CjdCas9-VPR fusion baseline rather than the minimized miniCAFE. Compare mini-DREAM and mini-DREAM compact to miniCAFE.

Supplementary Figure 21. Statistical significance measures for consistency and comparison against state-of-the-art VPR.

Supplementary Figure 21. Inclusion of Cas9-VP192 fusion for gene upregulation benchmarking alongside use in human fibroblasts and potential VP64-Cas9-VP64 inclusion as well to elucidate synergistic effects of VP64 inclusion.

Supplementary Figure 21. Show statistical significance between VP64 fusions and dCas9-VPR to substantiate the Supplementary Note 2 claim of fusion VP64 demonstrating a substantial expression increase within the OCT4 testbed.

Supplementary Figure 11e. Clarity on whether only bold and red font exist or if there is another category apparent.

Supplementary Figure 7. Clarity on statistical significance between MRTF-A and MRTF-B. Is there a significant decrease here between the two top choices?

Supplementary Figure 28a. Demonstrate consistent reads in flow cytometry data to eliminate potential subsampling bias in percentage viability of T-cells.

Reviewer #2:

Remarks to the Author:

Mahata et al. describe a suite of CRISPR-based transactivation (CRISPRa) systems using transcription activation domains (TADs) of mechanosensitive transcription factors (MTFs). Unlike most CRISPRa systems that rely on viral factors, the authors harness the robust natural transcriptional activation property of MTF proteins to induce programmable gene activation in mammalian cells. The engineering efforts described in the manuscript is one of the strongest in the literature. The highlights include: (1)

initial testing a panel of MTFs to identify the MSN/NMS combination as the most potent activator; (2) comparison of recruitment systems to endogenous genes (direct fusion, MCP, SAM); (3) miniaturization efforts of the system by using orthogonal CRISPR systems (including Type I and II CRISPR effector proteins) and non-CRISPR DNA binding platforms; (4) applications in cellular reprogramming of fibroblast to iPSCs, viral-based based delivery into primary cells, and packageable into AAV vectors.

Perhaps a notable comment of the study is whether the CRISPR-DREAM system is a major advancement in CRISPRa technologies. In many of the comparisons, the difference between CRISPR-DREAM, SAM, and dCas9-VPR are comparable. What could strengthen this point are any data to suggest that previous technologies that rely on viral transactivators are toxic. The authors point this out in the introduction but did not provide references: "...components derived from viral pathogens...are poorly tolerated in clinically important cell types, which could hamper biomedical or in vivo use". Although the authors' efforts to engineer more compact CRISPR-DREAM platforms is strong, the authors provide data in Figure 4 showing that the existing VPR activator performs similarly as MSN/NMS when used with TALE and ZF DNA binding platforms. A series of points to highlight the value of MSN/NMS over other existing tools would strengthen the manuscript.

Overall, the manuscript is well-written and it is clear the authors performed extensive validation of CRISPR-DREAM. It is appropriate for publication on Nature Methods and I recommend addressing the following major points:

1. The authors should expand on potential toxicity of CRISPR-DREAM, specifically on cell viability after overexpression of the MTFs. This can be done with live/dead cell staining 2-3 days post-transfection. The authors performed this with primary T cells (Supplementary Fig. 28), but it would be informative to perform this experiment with other cell lines. It is known that p300-based CRISPRa systems can be toxic in cells. A comparison of CRISPR-DREAM and other existing CRISPRa tools in terms of viability will be informative.
2. Almost all the data are from qPCR readouts from bulk transfected cells. I highly recommend performing a CRISPR-DREAM activation experiment, followed by antibody staining of the target protein and read out by flow cytometry (for example, CD34). This will provide a single cell comparison of the degree of activation of CRISPR-DREAM compared to dCas9-VPR.
3. It would be informative to measure time-dependent activation of genes of CRISPR-DREAM compared to SAM and dCas9-VPR. Do the dynamics match other technologies? How long does gene activation persist?
4. The authors could include descriptions on how the 7 different TADs were chosen. Potentially include a table/description of how these TAD domains are similar or different from each other

(structure/domain/sequence), specifically given that some of these failed to activate target genes while others worked well.

5. It appears that of all the proteins tested, only STAT1 fused with MRTF-A (DREAM components) enhances activation. Could the authors comment/suggest a hypothesis as to why this is? Supplement Figure 6: Were other STAT fusions tested against non-OCT4 genes as well? Did any of them work better? (Trying to understand if these activators are gene specific?)

6. CRISPR-DREAM components description/breakdown would be useful to see in Fig. 1b with the M/S/eN spelled out.

7. I would be informative to get an overview of how the MTFs function mechanistically to activate transcription. Direct recruitment of Mediator or a different pathway? Which epigenetic modifications are modified upon CRISPR-DREAM recruitment?

8. Discuss the caveats of the MTFs/CRISPR-DREAM platform. Are there any genes that CRISPR DREAM failed to activate? Potentially discuss why MS2 loops work better than direct fusion? Again, the authors could expand in the discussion section of how this system compares to other CRISPRa systems that have been developed (not be limited to SAM).

Minor comments

Supplement 10a, it is difficult to see guide labels on TTN gene

Include additional details on primary and secondary antibody concentrations/dilutions in materials and methods (for Westerns/IF)

Fig. 5: Hard to see scale bars on IF on main figure images

Reviewer #3:

Remarks to the Author:

The authors designed and engineered combinations of transactivation domains (TADs) derived from human transcription factors, termed the DREAM platform. They demonstrated its activation potential in conjunction with various CRISPR effectors, TALENs, and zinc finger proteins. The DREAM platform can also be effective when miniaturized by using smaller TADs (aka mini-DREAM), and compacted into a

single vector (aka mini-DREAM Compact). The authors then used a DREAM-dCas9 combination to reprogram human foreskin fibroblasts (HFFs) into induced pluripotent stem cells. Additionally, they demonstrated DREAM compatibility with viral delivery methods and viability of primary cells after transduction.

This manuscript is the culmination of considerable biological design, engineering, and experimentation. However, the proposed approach lacks evidence to demonstrate meaningful improvements over existing TAD technologies. We have some comments to improve the manuscript, but even if they were to be addressed, we believe the technology demonstrated here lacks the 'game-changing' impact necessary for publication in Nature Methods.

Main Points:

- 1) The authors claim in the main text that "eNRF2 displayed optimal potency" but supplementary figure 4 shows that the MCP-p65-HSF1 performed better than eNRF2 in all experiments. To avoid such misalignment between manuscript statements and graphical data representation, authors should be more precise in their wording. Additionally, the difference between the SAM system and MCP-p65-HSF1 is the lack of VP64, i.e. the removal of the one viral component from the SAM system. Considering how the authors make a case for the DREAM system by pointing out that it contains no viral components, it is important to make a head-to-head comparison between the DREAM and MCP-p65-HSF1 systems.
- 2) When claiming reprogramming of HFFs into iPSCs, authors rely on the gene expression markers and antibody stainings. The evidence should be further extended with experiments that confirm the true functional stemness of the created iPSCs. Authors should assess a differentiation potential of the created iPSCs. The authors acknowledge that DREAM resulted in fewer iPSC colonies than the existing dCas9-VP192 which diminishes its practical application value; perhaps, further DREAM-oriented protocol optimization could aid the efficiency.
- 3) Utility of mini-DREAM and mini-DREAM Compact could be strengthened if the authors provide stronger rationale and better use cases for these systems, especially given that their activation potential is dropping as the size is reduced. Arguments that miniature versions can help to avoid the limitations of AAV packaging capacity should be laid out more clearly. Mini-DREAM Compact size is $1578 \times 3 = 4734$ bp, so even if we exclude the necessary promoter sequence and highly desirable gRNA insert, this construct hits the upper limit of the AAV cargo capacity (appr. 4.7 kbp). For the split-vector system, authors should comment more on the practical application aspect - which technical differences arise from using multiple viral components, compared to only one?

Minor Points:

- 1) Authors should strengthen the literature-based support for the claim that viral TAD domains negatively impact the cells upon activation, and preferably support it with experimental data showing

that the DREAM system is explicitly devoid of such effects.

2) Cell transfections were conducted using the same ng amount of plasmids; it would be interesting to see a head-to-head comparison in which equal molar amounts of the DREAM and other vectors are tested.

3) The authors claim in the main text that “thousands of human transcription factors (TFs) and chromatin modifiers have yet to be systematically tested.” This statement seems inconsistent in light of previous studies, most notably PMID 33326746 and PMID 35016035.

4) Typo in the second paragraph of page 7 - remove “the” from “when fused the to dCas9”.

5) Bottom paragraph of page 10 refers to a Figure 5i which does not exist.

6) Supplementary figure 11C is confusing - more self explanatory visualization should be used.

7) The last panel to the right in supplementary figure 28A has a really small number of cells. Authors cannot draw any valid conclusion with such a low sample size.

8) Staining images in Figure 5 could be supplemented with zoomed-in examples which would depict the staining patterns on the cellular, rather than organoid level.

9) The use of the word “substantially” is misleading when referring to HBG1/HBG2 transcription differences. Supplementary figure 11D shows 2.79x which is relatively small when fold increase of either technology is in the thousand-fold range.

Author Rebuttal to Initial comments

Summary of Responses to Reviewers' concerns

We are sincerely grateful to each reviewer for their time, constructive feedback and thoughtful comments on our manuscript. We have addressed every comment from each reviewer in our point-by-point response below. In our revised manuscript we have included several new experiments (and associated data) and added new language to improve the clarity and accuracy of our work here. We believe that in addressing the comments from each Reviewer, we have strengthened both the findings and impacts of our work, hence we appreciate the thoughtful suggestions and productive critical concerns.

Notably, new data and significant changes in the main text and supplementary text are **annotated using yellow highlights** throughout. We have also updated source data and added new key references. Please also note that some Supplementary Figure numbering has been revised to incorporate new experiments/data. In this document, **new Figures /new Supplementary Figures** are denoted by bolded red font.

Point-by-point Responses to Reviewers' concerns

Reviewers' Comments:

Reviewer #1:

Remarks to the Author:

This work outlines the design and validation of a multipartite activation domain derived from human mechanosensitive transcription factors (MTFs) to be used in conjunction with CRISPR-dCas9. Validation of constructs involved a panel of cells and genetic targets (including noncoding genomic regulatory elements) as well as a multitude of DNA binding (zinc finger and TALENs) and CRISPR based (Type I, II, and V) platforms. Applications including iPSC fibroblast reprogramming and both dual and all-in-one (AIO) AAV packaging were demonstrated as proof of concept.

To the best of my knowledge, there have been no prior works investigating MTF TADs in the context of CRISPRa to this extent. In this work, comparisons against state of the art CRISPRa platforms were used throughout as effective positive controls in a variety of experiments. Final tripartite MSN, NMS, and eN3x9 modules saw relatively comparable and moderate increases in the majority of in vitro experiments performed, lending credibility to both novelty and feasibility as an alternative to current methods. Unique strengths of this method lie notably with increased cell viability in primary T cells and considerably improved performance with the SunTag and dCas12a platforms. Beneficial investigations included the evaluation of a compact construct involving minimized domains of the MTF TADs used in the full-size DREAM construct (similar to AIO miniCAFE development), methods for AIO delivery, and equivalent efficiency with readily modular integration of the multipartite domain to various existing modalities used for gene activation (ZFs and TALENs).

As of current implementation, this multipartite domain seemingly only holds true novel utility above existing platforms in the niche areas of aforementioned unique strengths. In particular, VP64, VP192, VPR, and p65-HSF1 (SAM) platforms still maintain comparable effectiveness (or in the case of VP192, seemingly improved performance in iPSC colony growth) in the majority of use cases explored and would likely see no inherent downsides in in vitro study. A specific benefit could be the easier integration into studies aiming for primarily human-derived platforms (like zinc fingers discussed in this study) or those concerned with immunogenicity/cytotoxicity

in general. The versatility of this domain to replace many of the state-of-the-art platforms without significant losses (unlike p65-HSF1 which appears to lose efficiency when fused) combined with the outlined framework for validation and minimization, facilitate utility and the prospects of a novel technology respectively, but begs an inquiry into how existing platforms would fair if subjected to the same evaluations and combinatorics. As these evaluations do not currently exist, there is, by absence, novelty of the actual multipartite domain developed.

This work provides a thorough analytical framework and progression to both development and validation of the tripartite domains, suggesting reasonable utility as an equivalent alternative to existing CRISPRa platforms. However, it is noted that the strong benefits of the framework (ie. facilitating overcoming notable challenges in this technology), T-cell viability, AIO AAV delivery, and a composition devoid of viral components derive their importance from the strong benefits implied for in vivo work, without explicit in vivo investigation here.

Overall, all claims and statements appear reasonable and are contained within the results shown; and there is novelty in the domains, but activity over and above existing domains is not observed, so broader utility (beyond niche applications around managing immunogenicity) remains uncertain.

RESPONSE 1.0: We are grateful to Reviewer 1 for their thoughtful comments and constructive input on our work here. Overall, we agree with and appreciate all suggestions and recommendations from Reviewer 1. Therefore, we performed all of the experiments suggested by Reviewer 1 and included these new data in our revised manuscript. In addition, we incorporated new numerical/statistical analyses and improvements upon technical clarity pursuant to the useful input provided by Reviewer 1.

Using rational design, focused combinatorial build cycles, and comprehensive testing, we developed the MTF-derived multipartite transactivation modules (MSN, NMS, and eN3x9) and showed that each of these modules are not only smaller than existing state-of-the-art CRISPRa platforms (the SAM module; VP64, p65 + HSF1, and the VPR module; VP64 + p65 + Rta), but also maintain superior or comparable transactivation properties across loci and cell types. Interestingly, we note that the prior CRISPRa platforms in many ways are iterative in that, they each build upon the early identifications of both the VP64 and p65 TADs. In contrast, the MSN, NMS, and eN3x9 modules are entirely novel in this respect and lack this historical lineage in the literature. We appreciate that this point was recognized by Reviewer 1 with respect to there being no prior work investigating MTF TADs in a general sense and in the context of CRISPRa.

In terms of optimizing the domain combinations for the existing state-of-the-art CRISPRa platforms (i.e., the SAM and VPR modules), VPR was in fact tested in this way and V-P-R was presented as the optimal configuration sequence for the VP64, p65, and Rta subunits (PMID: 25730490). For the SAM platform, we agree with the suggestion regarding possible optimization from Reviewer 1. To this end, we shuffled the positioning of the p65 and HSF1 subunits of the SAM system, and in fact observed a reduction in gene activation potency at two different target sites, with or without VP64 co-recruited via direct fusion to dCas9 (please see **new Supplementary Figs. 8e and 8f, respectively**). Therefore, we suspect that the p65/HSF1 ordering was also optimized previously (PMID: 25494202).

Minor Revisions and Recommendations:

Q1.1: Benchmark data vs MCP-p65-HSF instead of SAM could be desirable. Supplementary Figure 7 seems to demonstrate that MSN compared to p65-HSF1 without the viral VP64 fusion component of the SAM system is comparable to MSN and NMS. Potency of p65-HSF1 as an activation domain with recruitment may prove comparable to the tripartite fusion and would remain non-viral. Furthermore, potential changes to fusion conformations may see improvements similar to those seen with NMS and MNS.

respectively, but begs an inquiry into how existing platforms would fair if subjected to the same evaluations and combinatorics. As these evaluations do not currently exist, there is, by absence, novelty of the actual multipartite domain developed.

This work provides a thorough analytical framework and progression to both development and validation of the tripartite domains, suggesting reasonable utility as an equivalent alternative to existing CRISPRa platforms. However, it is noted that the strong benefits of the framework (ie. facilitating overcoming notable challenges in this technology), T-cell viability, AIO AAV delivery, and a composition devoid of viral components derive their importance from the strong benefits implied for in vivo work, without explicit in vivo investigation here.

Overall, all claims and statements appear reasonable and are contained within the results shown; and there is novelty in the domains, but activity over and above existing domains is not observed, so broader utility (beyond niche applications around managing immunogenicity) remains uncertain.

RESPONSE 1.0: We are grateful to Reviewer 1 for their thoughtful comments and constructive input on our work here. Overall, we agree with and appreciate all suggestions and recommendations from Reviewer 1. Therefore, we performed all of the experiments suggested by Reviewer 1 and included these new data in our revised manuscript. In addition, we incorporated new numerical/statistical analyses and improvements upon technical clarity pursuant to the useful input provided by Reviewer 1.

Using rational design, focused combinatorial build cycles, and comprehensive testing, we developed the MTF-derived multipartite transactivation modules (MSN, NMS, and eN3x9) and showed that each of these modules are not only smaller than existing state-of-the-art CRISPRa platforms (the SAM module; VP64, p65 + HSF1, and the VPR module; VP64 + p65 + Rta), but also maintain superior or comparable transactivation properties across loci and cell types. Interestingly, we note that the prior CRISPRa platforms in many ways are iterative in that, they each build upon the early identifications of both the VP64 and p65 TADs. In contrast, the MSN, NMS, and eN3x9 modules are entirely novel in this respect and lack this historical lineage in the literature. We appreciate that this point was recognized by Reviewer 1 with respect to there being no prior work investigating MTF TADs in a general sense and in the context of CRISPRa.

In terms of optimizing the domain combinations for the existing state-of-the-art CRISPRa

platforms (i.e., the SAM and VPR modules), VPR was in fact tested in this way and V-P-R was presented as the optimal configuration sequence for the VP64, p65, and Rta subunits (PMID: 25730490). For the SAM platform, we agree with the suggestion regarding possible optimization from Reviewer 1. To this end, we shuffled the positioning of the p65 and HSF1 subunits of the SAM system, and in fact observed a reduction in gene activation potency at two different target sites, with or without VP64 co-recruited via direct fusion to dCas9 (please see **new Supplementary Figs. 8e and 8f, respectively**). Therefore, we suspect that the p65/HSF1 ordering was also optimized previously (PMID: 25494202).

Minor Revisions and Recommendations:

Q1.1: Benchmark data vs MCP-p65-HSF instead of SAM could be desirable. Supplementary Figure 7 seems to demonstrate that MSN compared to p65-HSF1 without the viral VP64 fusion component of the SAM system is comparable to MSN and NMS. Potency of p65-HSF1 as an activation domain with recruitment may prove comparable to the tripartite fusion and would remain non-viral. Furthermore, potential changes to fusion conformations may see improvements similar to those seen with NMS and MNS.

Response 1.1: We appreciate these excellent and astute points from Reviewer 1. To address these points, we first performed a set of new experiments wherein we directly compared dCas9 + MCP-MSN and dCas9 + MCP-p65-HSF1 at 6 different target genes using pooled and single gRNAs (please see **new Supplementary Figs. 8c and 8d, respectively**). In each of these cases, dCas9 + MCP-MSN led to higher levels of gene activation than dCas9 + MCP-p65-HSF1. We also investigated whether potential changes to the fusion conformations of p65-HSF1 could augment potency, and observed no increases in relative potency, either in tandem with dCas9 or with dCas9-VP64 (please see **new Supplementary Figs. 8e and 8f, respectively**). In addition, we have revised our text to highlight these new comparisons and benchmarking data.

Q1.2: An overall suggestion is to indicate the fold change differences like in Supplementary Fig. 11d to better emphasize significant improvements where apparent across results and claims.

Response 1.2: We like this suggestion from Reviewer 1, however we found that trying to include all notations of statistical significance *and* relative fold increases at key results in the main figures resulted in distraction from the data. While we agree with this suggestion in principle, in practice it was visually unappealing. However, we fully agree with the goal of this suggestion, and have revised our language in respective figure legends to point the reader to our source data, wherein we have now included all fold changes with respect to significant improvements of key main text (and selected Supplementary) data panels.

Q1.3: Typo, Main-Page 5: sacas9 to sadcas9.

Response 1.3: Thank you for identifying this error, we have corrected this oversight and we have similarly modified our language regarding CjdCas9 as well.

Q1.4: Typo, Supplementary Figure 21a: ca9 to cas9.

Response 1.4: Again, thank you for spotting this error, we have corrected this oversight and appreciate the thoughtful attention to detail from Reviewer 1! Please note that some Supplementary Figure numbers have changed to incorporate new data (i.e., Supplementary Figure 21 is now Supplementary Figure 24).

Q1.5: Figure 3. Suggest a CjdCas9-VPR fusion baseline rather than the minimized miniCAFE. Compare mini-DREAM and mini-DREAM compact to miniCAFE.

Response 1.5: These are great suggestions. We have performed new experiments comparing CjdCas9-DREAM and CjdCas9-VPR as suggested by Reviewer 1, and these new data demonstrate that CjdCas9-DREAM is superior to CjdCas9-VPR at two testbed genes (please see **new Figure 3f**). We also compared mini-DREAM and mini-DREAM compact to miniCAFE and moved these data to the Supplement (please see **new Supplementary Fig. 23j and k**).

Q1.6: Supplementary Figure 21. Statistical significance measures for consistency and comparison against state-of-the-art VPR.

Response 1.6: We thank Reviewer 1 for this comment. For Supplementary Fig. 24a (revised numbering), we have added statistical analyses between NMS-dCas9-VP64 and dCas9-VPR. For all other panels in Supplementary Fig. 24, we have added statistical significance for key comparisons. We have also toned down our language to more clearly explain that NMS-dCas9-VP64 and dCas9-VPR are overall comparable in terms of gene activation potency in HEK293T cells, particularly in Supplementary Note 2. However, we maintain our point that NMS-dCas9-VP64 remains better expressed and is ~35% smaller than dCas9-VPR.

Q1.7: Supplementary Figure 21. Inclusion of Cas9-VP192 fusion for gene upregulation benchmarking alongside use in human fibroblasts and potential VP64-Cas9-VP64 inclusion as well to elucidate synergistic effects of VP64 inclusion.

Response 1.7: We appreciate this suggestion and we have included new data comparing NMS-dCas9-VP64, dCas9-VP192, and VP64-dCas9-VP64. To minimize any potential confounding batch effects, we have included these new data as **new Supplementary Fig. 24c**. These data indicate that NMS-dCas9-VP64 outperforms both dCas9-VP192 and VP64-dCas9-VP64 when targeted to the *OCT4* promoter in HEK293T cells and they also demonstrate that dCas9-VP192 is superior to VP64-dCas9-VP64 in these contexts, which is consistent with a recent report (PMID: 33401508). We have also included the sizes of respective fusions components of dCas9-VP192 and VP64-dCas9-VP64 in **new Supplemental Fig. 24d**.

Q1.8: Supplementary Figure 21. Show statistical significance between VP64 fusions and dCas9- VPR to substantiate the Supplementary Note 2 claim of fusion VP64 demonstrating a substantial expression increase within the OCT4 testbed.

Response 1.8: We thank Reviewer 1 for this comment, and in addition to the modifications pursuant to Q1.6 above (i.e., toning down our language), we have also included statistical analyses for associated comparisons in Supplementary Fig. 24 (revised numbering).

Q1.9: Supplementary Figure 11e. Clarity on whether only bold and red font exist or if there is another category apparent.

Response 1.9: We are grateful to Reviewer 1 for identifying this lack of clarity. We have modified the legend of Supplementary Figure 13d (revised numbering) to correct our oversight.

Q1.10: Supplementary Figure 7. Clarity on statistical significance between MRTF-A and MRTF-B. Is there a significant decrease here between the two top choices?

Response 1.10: We believe that Reviewer 1 is referring to MSN vs. NMS in Supplementary Fig. 8b (revised numbering), and in this case there is no significant difference in potency between dCas9 + MSN and dCas9 + NMS (but we apologize if we are misunderstanding the point of Reviewer 1). Nevertheless, we selected MSN as the lead variant because quantitatively (although not to a statistically significant level) it was superior to the NMS architecture at the OCT4 testbed locus.

Q1.11: Supplementary Figure 28a. Demonstrate consistent reads in flow cytometry data to eliminate potential subsampling bias in percentage viability of T-cells.

Response 1.11: We appreciate Reviewer 1 raising this important concern regarding potential subsampling bias. We also appreciate this opportunity to clarify our gating strategy, rationale, and experimental readouts. All effector fusions in this experiment also encoded C-terminal T2A-EGFP, thus EGFP served as a readout for effector positive cells. To measure apoptosis and membrane integrity in EGFP positive (and hence effector positive) T cells, we first gated to remove any debris (SSC-A vs. FSC-A) and then gated for single T cells (FSC-A Vs. FSC-H; please see **new Supplementary Fig. 31a**; revised numbering). Spectral unmixing was performed to minimize fluorometric spillover as per manufacturer instruction; specifically, EGFP signals of $<1E-3$ (FSC- A vs. EGFP-A) were excluded. We next sorted EGFP positive T cells (FSC-A vs. EGFP-A; please see **new Supplementary Fig. 31b**) to focus our cytotoxicity analyses on cells that were transduced with respective effectors. We then measured Annexin V-PE (PE-A; a readout for early apoptosis) and 7-AAD (7AAD-A; a readout for late apoptosis/necrotic cells) intensities in these resultant cell populations (Supplementary Fig. 31c).

We selected this methodology and gating strategy to analyze the cells that we were certain contained transcriptional effector fusions (i.e., EGFP positive cells). Importantly, across all these experiments we measured the same total numbers of primary T cells (~3E4 T cells per

experimental replicate/condition). Further, all viruses were packaged, titered, and transduced together with all respective effector constructs at the same MOIs across conditions. Thus, taken together, we are highly confident of two observations:

- i) There is differential expression of these tested effector constructs (as measured via the EGFP proxy) in primary T cells.
- ii) In primary T cells that actively express these respective effectors/EGFP fusions, MCP- VPR, MCP-MSN, and MCP-NMS are more cytotoxic than MCP-eN3X9.

That said, our experiments do not allow us to rule out the possibility that MCP-VPR, MCP-MSN, and MCP-NMS do not lead to impaired lentiviral entry relative to MCP-eN3X9 in primary T cells, nor whether the tested effectors caused cell death prior to assay. In addition to adding our gating strategy data to Supplementary Fig. 31, we have also included all relevant cytometry data for primary T cells (including control data) as source data. Finally, we have revised our main and supplementary text throughout to add clarity to this experiment and associated data.

Reviewer #2:

Remarks to the Author:

Mahata et al. describe a suite of CRISPR-based transactivation (CRISPRa) systems using transcription activation domains (TADs) of mechanosensitive transcription factors (MTFs). Unlike most CRISPRa systems that rely on viral factors, the authors harness the robust natural transcriptional activation property of MTF proteins to induce programmable gene activation in mammalian cells. The engineering efforts described in the manuscript is one of the strongest in the literature. The highlights include: (1) initial testing a panel of MTFs to identify the MSN/NMS combination as the most potent activator; (2) comparison of recruitment systems to endogenous genes (direct fusion, MCP, SAM); (3) miniaturization efforts of the system by using orthogonal CRISPR systems (including Type I and II CRISPR effector proteins) and non-CRISPR DNA binding platforms; (4) applications in cellular reprogramming of fibroblast to iPSCs, viral-based based delivery into primary cells, and packageable into AAV vectors.

Perhaps a notable comment of the study is whether the CRISPR-DREAM system is a major advancement in CRISPRa technologies. In many of the comparisons, the difference between CRISPR-DREAM, SAM, and dCas9-VPR are comparable. What could strengthen this point are any data to suggest that previous technologies that rely on viral transactivators are toxic. The authors point this out in the introduction but did not provide references: "...components derived from viral pathogens...are poorly tolerated in clinically important cell types, which could hamper biomedical or in vivo use". Although the authors' efforts to engineer more compact CRISPR-DREAM platforms is strong, the authors provide data in Figure 4 showing that the existing VPR activator performs similarly as MSN/NMS when used with TALE and ZF DNA binding platforms. A series of points to highlight the value of MSN/NMS over other existing tools would strengthen the manuscript.

Overall, the manuscript is well-written and it is clear the authors performed extensive validation of CRISPR-DREAM. It is appropriate for publication on Nature Methods and I recommend addressing the following major points:

RESPONSE 2.0: We are extremely grateful to Reviewer 2 for their constructive and thoughtful feedback. We are confident that the transactivation modules developed here from human MTFs and the CRISPR-DREAM platform variants are major, and importantly useful, advancements for 3 key reasons:

- i. These new modules are relatively compact, especially compared to competing/existing technologies. This has enabled us to create a potent all-in-one AAV CRISPRa platform that we suspect will be broadly useful to the community.
- ii. These modules are functionally quite useful, because we have shown that they are immediately compatible with all programmable DNA binding.
- iii. These modules are also functionally quite useful because we have shown that they are both well-tolerated and efficacious in a wide array of mammalian cells, including clinical-grade primary human cells.

In addition, from a more methodological perspective, our work here provides a useful reference for future synthetic transcription factor benchmarking and development studies. This could extend to future iterations wherein the MSN, NMS, and/or eN3x9 TADs (or other 9aa segments) could be dovetailed with existing or future tools/toolkits.

We regret that we mistakenly did not include key references regarding the potential for toxicity and intolerance of components derived from viral pathogens. In our revised manuscript, we have included new references that highlight the toxicity of certain existing technologies, and simultaneously we have toned down our language surrounding the direct toxicity of viral components, which to our knowledge has not been conclusively demonstrated.

Q2.1: The authors should expand on potential toxicity of CRISPR-DREAM, specifically on cell viability after overexpression of the MTFs. This can be done with live/dead cell staining 2-3 days post-transfection. The authors performed this with primary T cells (Supplementary Fig. 28), but it would be informative to perform this experiment with other cell lines. It is known that p300-based CRISPRa systems can be toxic in cells. A comparison of CRISPR-DREAM and other existing CRISPRa tools in terms of viability will be informative.

Response 2.1: We appreciate this excellent suggestion from Reviewer 2. To address this concern and to expand upon the potential toxicity/effects upon cell viability between CRISPR-DREAM and other existing CRISPRa tools, we performed live/dead cell staining 2 days after CRISPR-DREAM, the SAM system, dCas9-VPR, or dCas9-p300 were transiently transfected into either U2OS or HEK293T human cell lines (targeting each to the *HBG1* promoter), as suggested by Reviewer 2. Here, we used 7AAD staining and microscopy to measure cytotoxicity in these adherent cell lines. We first tried to assay toxicity using flow cytometry but observed that the requisite processing was detrimental to these cell death measurements in

adherent cells, which we attribute to the presence of residual trypsin, and/or long periods in suspension, and/or other unknown factors. Nevertheless, our new data reveal that in transformed cell lines, the relative levels of cytotoxicity appear to be more equivalent among the CRISPRa tools tested (please see **new Supplementary Fig. 32**). However, in HEK293T cells, dCas9-p300 did lead to subtle morphological changes in culture, and to measurably higher levels of cell death.

Q2.2: Almost all the data are from qPCR readouts from bulk transfected cells. I highly recommend performing a CRISPR-DREAM activation experiment, followed by antibody staining of the target protein and read out by flow cytometry (for example, CD34). This will provide a single cell comparison of the degree of activation of CRISPR-DREAM compared to dCas9-VPR.

Response 2.2: This is an exciting and incisive suggestion from Reviewer 2. To address this concern, we performed two sets of experiments. First, as proposed by the reviewer, we transfected dCas9, CRISPR-DREAM, the SAM system, and dCas9-VPR (without any fluorescent tags) into HEK293T cells and targeted each platform to the *CD34* promoter. We then used flow cytometry to measure the resulting CD34 cell-surface protein levels. After gating for single cells, we observed that both the CRISPR-DREAM and SAM systems result in similar CD34 positive proportions and mean fluorescence intensities (MFIs; please see **new Supplementary Fig. 10a-c**). Interestingly, dCas9-VPR performed significantly lower in both metrics.

In the second experiment relevant to addressing this concern, we utilized MCP recruitment for all effectors (MCP-MSN, MCP-P65-HSF1, and MCP-VPR), and each effector fusion also encoded a T2A EGFP fluorophore, which enabled us to more faithfully measure how increased CD34 cell-surface levels correlated with respective effector expression on a single cell level. These data demonstrated that CRISPR-DREAM resulted in a greater proportion of EGFP positive and EGFP/CD34 double positive cells (please see **new Supplementary Fig. 10d-h**). In addition, we observed that CRISPR-DREAM resulted in the greatest absolute CD34 levels (please see **new Supplementary Fig. 10i-k**). Therefore, we are confident that CRISPR-DREAM is more potent than dCas9-VPR both when measured using QPCR in bulk cell populations and using flow cytometry analyses of single cells.

Q2.3: It would be informative to measure time-dependent activation of genes of CRISPR-DREAM compared to SAM and dCas9-VPR. Do the dynamics match other technologies? How long does gene activation persist?

Response 2.3: Again, we appreciate this excellent suggestion from Reviewer 2. To address this interesting point, we measured the time-dependent activation of *HBG1* and *SBNO2* genes in response to targeting with CRISPR-DREAM, the SAM system, and dCas9-VPR over 20 and 12 days, respectively (please see **new Supplementary Fig. 14**). Our results demonstrate that at these loci in HEK293T cells, although CRISPR-DREAM maintains superior efficacy over the SAM and dCas9-VPR systems, all platforms can maintain statistically significant levels of gene activation over extended time periods – even in rapidly dividing and repeatedly passaged

HEK293T cells. Of note, *SBNO2* data collection ended at 12 days due to high levels of experimental variance; likely because the basal Ct values for *SBNO2* were extremely low/undetectable, and because Ct values for all CRISPRa experiments targeting *SBNO2* yielded Ct values near ~35 cycles.

Q2.4: The authors could include descriptions on how the 7 different TADs were chosen. Potentially include a table/description of how these TAD domains are similar or different from each other (structure/domain/sequence), specifically given that some of these failed to activate target genes while others worked well.

Response 2.4: This is a good point that we believe could be useful for future tool design/optimization. At first, our designs were largely rationally guided by the literature and on mechanosensitive transcription factors (MTFs) because “*The robust, highly orchestrated, and relatively ubiquitous gene regulatory effects of these classes of human MTFs make them excellent potential sources of new non-viral TADs that could be leveraged as components of engineered CRISPRa systems and/or other synthetic gene activation platforms*”. We first selected 7 mechanosensitive TADs chosen from 6 different transcription factors which were known to play key gene regulatory roles in two distinct signaling cascades. YAP and TAZ both contain established C-Terminal TADs and are paralogous transcriptional co-activators within the Hippo signaling cascade that exert their transcriptional activity by binding to TEAD DNA binding proteins. We also selected a mutant YAP-S397A TAD, which had been shown to harbor hypertranscriptional activity. MYOCD, MRTF-A, MRTF-B, and SRF are transcriptional co-activators within the MRTF-MYOCD-SRF signaling cascade. We selected TAD domains from these proteins based upon prior literature indicating TAD boundaries.

To more directly address this suggestion from Reviewer 2, we have performed clustal omega alignments of the YAP and TAZ TADs (please see **new Supplementary Fig. 6a**), the MRTF-A, MRTF-B and MYOCD TADs (please see **new Supplementary Fig. 6c**), STAT1-6 TADs (please see **new Supplementary Fig. 6d**), and the Neh4 and Neh5 domains of NRF2 along with the eNRF2 TAD (please see **new Supplementary Fig. 6e**) to highlight potential similarities at the amino acid level between these 15 different putative TADs. To further understand how sequence composition within TADs might impact efficacy, we also performed 9aa TAD searches (PMIDs: 17467954 and 25564305) for each of these 15 protein segments. To our surprise every single TAD that showed significant activity when recruited via dCas9 harbored at least one 9aa TAD with 100% match to the algorithmic prediction (please see **new Supplementary Fig. 6f**). Therefore, we consider this algorithm to have substantial predictive potential.

Q2.5: It appears that of all the proteins tested, only STAT1 fused with MRTF-A (DREAM components) enhances activation. Could the authors comment/suggest a hypothesis as to why this is? Supplement Figure 6: Were other STAT fusions tested against non-OCT4 genes as well? Did any of them work better? (Trying to understand if these activators are gene specific?)

Response 2.5: Great question. Our experiments in Supplementary Fig. 7c (revised numbering)

showed that all (24 in total) MRTF-A/MRTF-B and STAT (1 through 6) bipartite fusions led to enhanced *OCT4* activation beyond the levels observed by MRTF-A or MRTF-B alone. However, the MRTF-A-STAT1 fusion showed the highest relative activation among all the 24 fusions tested. Therefore, we selected MRTF-A-STAT1 for subsequent deployment based on this relative potency, and due to its smaller size compared to other tested fusions. Although every one of these 24 different fusions was not evaluated at other testbed loci, we did test MRTF-A-STAT1 fusions at two other testbed genes (please see **new Supplementary Figs. 7d and 7e**). In each of these cases, we found enhanced activation as compared to only MRTF-A. We believe that the enhanced functions of MRTF-STAT fusion proteins relative to MRTFs alone are universal, however it could be possible that other fusions could perform better at other testbed loci. We feel that this is an exciting component of future work.

Q2.6: CRISPR-DREAM components description/breakdown would be useful to see in Fig. 1b with the M/S/eN spelled out.

Response 2.6: We appreciate this suggestion and have modified Fig. 1b accordingly.

Q2.7: I would be informative to get an overview of how the MTFs function mechanistically to activate transcription. Direct recruitment of Mediator or a different pathway? Which epigenetic modifications are modified upon CRISPR-DREAM recruitment?

Response 2.7: MTFs are a special class of transcription factors/coactivators harboring TADs and mechano-sensation/ligand sensitive regulatory domains that can harbor a spectrum of posttranslational modifications (i.e., phosphorylation, ubiquitination, etc.). These PTMs can in turn interact with other cellular proteins and these regulatory domains are thought to play a crucial role in sequestering these MTFs in the cytosol, and then rapidly translocating them to the nucleus upon stimulation – which ultimately results in robust transcriptional responses (please see PMIDs: 22863277 and 15292266).

Mechanistically, MTFs can induce transcription via interactions with H3K27 histone acetyl transferases (e.g., p300 and CBP), H3K16 histone acetyl transferases (e.g., TIP60), H3K4 methylation modulators (e.g., SET1), mediator proteins (e.g., MED16), other transcriptional regulatory proteins, as well as RNA Pol II (please see PMIDs: 19214187, 11683914, 26439301, 17599918 and 25159611). We have provided a schematic depiction of some of these important positive modulators of transcription observed to interact with MRTF-A, STAT1, and NRF2 (please see **new Supplementary Fig. 34a**).

Based on these previous studies, we hypothesized that H3K27 acetylation and H3K4 methylation could possibly be upregulated upon engagement of the CRISPR-DREAM effector MSN fusion protein at human promoters. To test this hypothesis, we performed CUT&RUN-QPCR to analyze H3K27ac and H3K4me3 levels when CRISPR-DREAM was targeted to the *HBG1* promoter. We also included dCas9 and the known epigenome editor dCas9-p300 (which modulates H3K27ac levels) as controls. Our new data (please see **new Supplementary Figs. 34b and c**) demonstrate that CRISPR-DREAM (harboring the MSN effector) significantly

upregulated H3K27ac and H3K4me. We have revised our text and added new references to highlight these new mechanistic insights and thank the reviewer for the suggestion.

Q2.8: Discuss the caveats of the MTFs/CRISPR-DREAM platform. Are there any genes that CRISPR DREAM failed to activate? Potentially discuss why MS2 loops work better than direct fusion? Again, the authors could expand in the discussion section of how this system compares to other CRISPRa systems that have been developed (not be limited to SAM).

Response 2.8: We appreciate these important points and we have expanded our discussion section to address them. With respect to caveats to MTFs and CRISPR-DREAM, we point out that further protein engineering might yield even stronger and durable CRISPRa platforms. We also note that although we tried to activate genes related to cardiac reprogramming with both CRISPR-DREAM and the SAM system in HEK293T cells (using gRNAs from PMID: 32082976 targeting *MEF2C*, *MEIS1*, and *GATA4*; please see **new Supplementary Fig. 35**), we failed to achieve significant gene induction. Our working hypothesis is that this lack of activation is due to the high basal levels of gene expression at these loci (*MEF2C* Ct value ~22; *MEIS1* Ct value ~22; and *GATA4* Ct value ~22) mRNA expression, and not the lack of platform efficacy *per se*.

In terms of the apparent MS2-related fusion architecture gene activation superiority, we suspect that MS2 loops may offer increased flexibility and/or reduced degradation of dCas9-based CRISPRa fusions. However, we have recently observed that effector, gRNA-targeted site, and cell type, can all impact CRISPRa efficacy (PMID: 35849129), suggesting that much more work is needed to clearly establish the rules for the efficacy of particular CRISPRa platforms, and CRISPRa more generally. One exciting area that could be useful in this regard is high-throughput screening of effectors and loci. However, while exciting and urgently needed, we feel these efforts are beyond the scope of this current work.

Minor comments

Q2.9: Supplement 10a, it is difficult to see guide labels on TTN gene

Response 2.9: Understood and addressed. We have revised Supplementary Fig. 12 (revised numbering).

Q2.10: Include additional details on primary and secondary antibody concentrations/dilutions in materials and methods (for Westerns/IF)

Response 2.10: We have corrected these oversights in our materials and methods section.

Q2.11: Fig. 5: Hard to see scale bars on IF on main figure images.

Response 2.11: Understood and addressed. We have revised the scale bars in Fig. 5 and in Supplementary Fig. 29j (revised numbering).

Reviewer #3:

Remarks to the Author:

The authors designed and engineered combinations of transactivation domains (TADs) derived from human transcription factors, termed the DREAM platform. They demonstrated its activation potential in conjunction with various CRISPR effectors, TALENs, and zinc finger proteins. The DREAM platform can also be effective when miniaturized by using smaller TADs (aka mini- DREAM), and compacted into a single vector (aka mini-DREAM Compact). The authors then used a DREAM-dCas9 combination to reprogram human foreskin fibroblasts (HFFs) into induced pluripotent stem cells. Additionally, they demonstrated DREAM compatibility with viral delivery methods and viability of primary cells after transduction.

This manuscript is the culmination of considerable biological design, engineering, and experimentation. However, the proposed approach lacks evidence to demonstrate meaningful improvements over existing TAD technologies. We have some comments to improve the manuscript, but even if they were to be addressed, we believe the technology demonstrated here lacks the 'game-changing' impact necessary for publication in Nature Methods.

RESPONSE 3.0: We are grateful to Reviewer 3 for their feedback, positive comments, and time in reviewing our manuscript. We regret that we did not more clearly specify the meaningful improvements and impacts associated with our new technologies developed here. Since 2015 there have been no new robust TADs identified for use with CRISPRa at endogenous human loci, and to date, nearly all CRISPRa studies rely upon a small handful of TADs (i.e., VP64, p300, p65, HSF1 or Rta). Our study here expands this list to new human TADs that are predictable, portable, potent, and compact. This new source material, and the "all in one" AAV CRISPRa platforms will be powerful components of engineered gene activation moving forward.

In addition, from a more methodological standpoint, our work here provides a useful reference for future synthetic transcription factor benchmarking and development studies. This could extend to future iterations wherein the MSN, NMS, and/or eN3x9 TADs (or other 9aa segments) could be dovetailed with existing or future tools/toolkits.

Main Points:

Q3.1: The authors claim in the main text that "eNRF2 displayed optimal potency" but supplementary figure 4 shows that the MCP-p65-HSF1 performed better than eNRF2 in all experiments. To avoid such misalignment between manuscript statements and graphical data representation, authors should be more precise in their wording. Additionally, the difference between the SAM system and MCP-p65-HSF1 is the lack of VP64, i.e. the removal of the one viral component from the SAM system. Considering how the authors make a case for the DREAM system by pointing out that it contains no viral components, it is important to make a head-to-head comparison between the DREAM and MCP-p65-HSF1 systems.

Response 3.1: We appreciate this comment from Reviewer 3 and regret that our wording was not more clear. We intended to explain that eNRF2 displayed optimal potency in the MS2 format

as compared to direct fusion or SunTag architectures (please see Supplementary Fig. 4a and b). MCP-p65-HSF1 was used as a positive control in related experiments (please see Supplementary Fig. 4c-e) and our claims are not intended to suggest that eNRF2 is superior to MCP-p65-HSF1. We have modified our language to reflect this important concern and to improve clarity.

We agree that head-to-head comparisons between the DREAM and MCP-p65-HSF1 systems are warranted, and we have performed a set of new experiments wherein we directly compared dCas9 + MCP-MSN and dCas9 + MCP-p65-HSF1 at 6 different target genes using both pooled and single gRNAs (please see **new Supplementary Figs. 8c and 8d**). In each of these cases, dCas9 + MCP-MSN led to higher levels of gene activation than dCas9 + MCP-p65-HSF1. We have revised our text to highlight these new comparative data and appreciate the suggestion from Reviewer 3.

Q3.2: When claiming reprogramming of HFFs into iPSCs, authors rely on the gene expression markers and antibody stainings. The evidence should be further extended with experiments that confirm the true functional stemness of the created iPSCs. Authors should assess a differentiation potential of the created iPSCs. The authors acknowledge that DREAM resulted in fewer iPSC colonies than the existing dCas9-VP192 which diminishes its practical application value; perhaps, further DREAM-oriented protocol optimization could aid the efficiency.

Response 3.2: The stemness of iPSCs in this testbed assay has been established (please see PMID: 29980666; reference #17 in our manuscript). We agree that a DREAM optimized protocol could aid improving reprogramming efficiency, however this assay simply serves as a proof of concept demonstrating that DREAM is functional beyond simple transcriptional activation.

Q3.3: Utility of mini-DREAM and mini-DREAM Compact could be strengthened if the authors provide stronger rationale and better use cases for these systems, especially given that their activation potential is dropping as the size is reduced. Arguments that miniature versions can help to avoid the limitations of AAV packaging capacity should be laid out more clearly. Mini-DREAM Compact size is $1578 \times 3 = 4734$ bp, so even if we exclude the necessary promoter sequence and highly desirable gRNA insert, this construct hits the upper limit of the AAV cargo capacity (appr. 4.7 kbp). For the split-vector system, authors should comment more on the practical application aspect - which technical differences arise from using multiple viral components, compared to only one?

Response 3.3: Again, we regret that our language was not clear on this point. We do not propose that mini-DREAM nor mini-DREAM compact will avoid the size limitations imposed by AAV vectors, only that our data show that the components of the CRISPR-DREAM system can be minimized to fit within a single vector delivery framework while retaining functionality.

With respect to practical application of the split-vector mini-DREAM system, we have added new experimental data showcasing the use of this technology to build a progesterone-producing HEK293T cell factory (please see **new Figs. 3k – n**).

Minor Points:

Q3.4: Authors should strengthen the literature-based support for the claim that viral TAD domains negatively impact the cells upon activation, and preferably support it with experimental data showing that the DREAM system is explicitly devoid of such effects.

Response 3.4: Recent work cited in our manuscript has provided evidence in support of these claims. For example, it has been shown that high levels of expression of dCas9-VPR and the dCas9-SAM system can be lethal in *Drosophila* (PMID: 28808002). Furthermore, when dCas9-VPR or dCas9-VP64 were injected into the yolk sacs of one-cell stage zebrafish embryos, both CRISPRa fusions displayed toxicity with or without guide RNAs (PMID: 34406040). Studies have also shown that the mice with the constitutional expression of dCas9-VPR are perinatally lethal, indicating that the dCas9-VPR is also toxic in mice (PMID: 32445790). These authors went on to suggest that the dCas9-VPR itself was toxic, and that its expression in inhibitory neurons aggravates epileptic seizures and sudden death in *Scn1a*-haplodeficient mice when expressed at the embryonic or early postnatal stages. Furthermore, recent *in vivo* CRISPRa studies have suggested that for clinical translation, CRISPRa tools should be compositionally optimized, and evaluated for toxicity in animal models (PMID: 31611701), which although beyond the scope of our work here, is an important and exciting area for future research effort.

Nevertheless, our data clearly demonstrate that, in primary T cells, the VPR domain is relatively poorly expressed and more toxic to cells than NMS, MSN, or eN3X9 (please see Supplementary Fig. 31). To more directly address this comment, we have added increased discussion regarding viral TADs and importantly, we have revised our language to be more conservative and to highlight the need for future *in vivo* toxicity evaluations, which again, are beyond the scope of our current work.

Q3.5: Cell transfections were conducted using the same ng amount of plasmids; it would be interesting to see a head-to-head comparison in which equal molar amounts of the DREAM and other vectors are tested.

Response 3.5: The components used in the DREAM system contain plasmids encoding dCas9 (12,878 bp) and MCP-MSN (11,340 bp). The components used in the SAM system contain plasmids encoding dCas9-VP64 (13,034 bp) and MCP-p65-HSF1 (11,409 bp). Throughout our experiments we used 187.5ng of each component in a 24 well plate. To address this concern, as per the suggestion from Reviewer 3, we performed equimolar experiments wherein, since the SAM system contains larger effectors, we used 189.77ng of dCas9-VP64 and 186.64ng of MCP- p65-HSF1. Our new data demonstrate similarly that there are no detectable differences in these experiments between using equal masses or equal molar concentrations of DNA using pools or single gRNAs (please see **new Supplementary Figs. 9n and o and Supplementary Figs. 11j and k**, respectively).

Q3.6: The authors claim in the main text that “thousands of human transcription factors (TFs) and chromatin modifiers have yet to be systematically tested.” This statement seems

inconsistent in light of previous studies, most notably PMID 33326746 and PMID 35016035.

Response 3.6: We appreciate this important and shrewd comment from Reviewer 3. Each of these valuable articles tested and validated thousands of transcription factors/domains using innovative high throughput methods. However, these studies selected only one type of platform to recruit and validate these transcriptional activation domains, which may not represent the optimal recruitment strategy. For instance, in PMID: 35016035, the authors solely used a chemically induced dimerization method to recruit entire repertoire of the human ORFeome, which may be an optimal recruitment strategy for many, but not all, ORFs. Similarly, in PMID 33326746, the authors used the rTetR DNA-binding domain to recruit size-restricted DNA fragments (all fragments were 300 nucleotides each) using the Pfam database.

Interestingly, both of these high-throughput strategies relied upon synthetic reporter systems, and surprisingly, neither identified any MTF TADs. Further, neither study benchmarked their identified transactivator hits against strong existing transcriptional activators (i.e., VP64, VPR, p65, HSF1, VPR) at any natural endogenous sites (for example, please check the “limitations of the study section” of PMID 33326746). Here, in contrast to these studies we; **i)** focused on using defined human mechanosensitive TADs, **ii)** we tested these TADs at several diverse endogenous loci, and **iii)** “systematically” tested these TADs and combinations of TADs using four different CRISPR-based recruitment architectures (and TALE- and ZF-mediated recruitment). However, to be clearer and more conscientious, we have modified our language to reflect these important and valuable prior findings.

Q3.7: Typo in the second paragraph of page 7 - remove “the” from “when fused the to dCas9”.

Response 3.7: We have corrected this error and thank Reviewer 3 for identifying this mistake.

Q3.8: Bottom paragraph of page 10 refers to a Figure 5i which does not exist.

Response 3.8: We thank Reviewer 3 for identifying this typographical mistake. The sentence has been corrected to “Figure 6h”.

Q3.9: Supplementary figure 11C is confusing - more self explanatory visualization should be used.

Response 3.9: We agree that this panel is confusing, and we have therefore removed it from Supplementary Fig. 14 (revised numbering).

Q3.10: The last panel to the right in supplementary figure 28A has a really small number of cells. Authors cannot draw any valid conclusion with such a low sample size.

Response 3.10: This is an excellent point, which was also noted by Reviewer 1. Again, we appreciate this important concern and appreciate the opportunity to clarify our gating strategy, rationale, and experimental readouts. As mentioned above in response 1.11, to measure apoptosis and membrane integrity in EGFP positive T cells, we first gated to remove any cell

debris, and then gated for single T cells (please see **new Supplementary Fig. 31a**). Spectral unmixing was then performed as per manufacturer instruction to minimize fluorometric spillover; specifically, EGFP signals of $< 1E-3$ (FSC-A vs. EGFP-A) were excluded. We next gated for EGFP positive T cells to focus our cytotoxicity analyses on cells that were transduced with respective effectors (please see **new Supplementary Fig. 31b**). We then measured Annexin V-PE and 7- AAD intensities (PE-A vs. 7AAD-A) in these resultant cell populations (Supplemental Fig. 31c).

We chose this methodology and gating strategy to analyze those cells that we were certain contained transcriptional effector fusions (i.e., EGFP positive cells). Importantly, across all these experiments we measured the same number of primary T cells ($\sim 3E4$ T cells per condition). Further, all viruses were packaged, titered, and transduced together with all respective effector constructs at the same MOIs across conditions. Thus together, we are highly confident of two things: i) There is differential expression of these tested effector constructs (as measured via the direct EGFP proxy) in primary T cells; ii) in primary T cells that actively express these respective effectors/EGFP fusions MCP-VPR, MCP-MSN, and MCP-NMS are more cytotoxic than MCP- eN3X9. That said, our experiments do not allow us to rule out the possibility that MCP-VPR, MCP- MSN, and MCP-NMS do not lead to impaired lentiviral entry relative to MCP-eN3X9 in primary T cells, nor whether the tested effectors caused cell death prior to assay.

In addition to adding our gating strategy data to Supplementary Fig. 31, we have also included all relevant cytometry data for primary T cells (including control data) as source data. Finally, we have revised our main and supplementary text throughout to add clarity to this experiment and associated data.

Q3.11: Staining images in Figure 5 could be supplemented with zoomed-in examples which would depict the staining patterns on the cellular, rather than organoid level.

Response 3.11: We appreciate this concern, but these iPSC colonies are compact masses, which lack clear boundaries between each cell/layer of cells. Our work here was simply a proof of concept to functionally benchmark dCas9-NMS against dCas9-VP192.

Q3.12: The use of the word “substantially” is misleading when referring to HBG1/HBG2 transcription differences. Supplementary figure 11D shows 2.79x which is relatively small when fold increase of either technology is in the thousand-fold range.

Response 3.12: We appreciate this concern from Reviewer 3 regarding the substantial fold change differences between the DREAM and SAM platforms in this experiment. We have revised our language to reflect our agreement with this concern.

Decision Letter, first revision:

Dear Isaac,

Thank you for submitting your revised manuscript "Compact engineered human mechanosensitive transactivation modules enable potent and versatile synthetic transcriptional control" (NMETH-A51570A). It has now been seen by the original referees and their comments are below. The reviewers find that the paper has improved in revision, and therefore we'll be happy in principle to publish it in Nature Methods, pending minor revisions to satisfy the referees' final requests and to comply with our editorial and formatting guidelines.

TRANSPARENT PEER REVIEW

ORCID

Sincerely,
Madhura

Madhura Mukhopadhyay, PhD
Senior Editor
Nature Methods

Reviewer #1 (Remarks to the Author):

The authors have reasonably addressed my comments.

Reviewer #2 (Remarks to the Author):

The authors addressed all of our comments and we feel the manuscript is appropriate for publication on Nature Methods.

Reviewer #3 (Remarks to the Author):

We appreciate that the authors have generally been responsive to each of our individual comments, but our prior judgement, but in the absence of substantially new demonstrations of experimental impact, our prior assessment remains the same: "we believe the technology demonstrated here lacks the 'game-changing' impact necessary for publication." We have no doubt that these activator domains function as described, but we are highly skeptical that the field will start to move towards these new TADs en masse, in which case this seems better suited to a more specialized journal.

Author Rebuttal, first revision:

Summary of Responses to Reviewers' concerns

We sincerely appreciate each reviewer for their time, constructive feedback and thoughtful

comments on our manuscript. We have addressed each final comment from each reviewer in our point-by-point response below.

Sincerely,
Isaac Hilton

Point-by-point Responses to Reviewers' concerns

Reviewers' Comments:

Reviewer #1:

Remarks to the Author:

The authors have reasonably addressed my comments.

RESPONSE: We are grateful to Reviewer 1 for their supportive and constructive comments and for taking the time to carefully review our manuscript.

Reviewer #2:

Remarks to the Author:

The authors addressed all of our comments and we feel the manuscript is appropriate for publication on Nature Methods.

RESPONSE: We are grateful to Reviewer #2 for their supportive comments and helpful suggestions throughout the review process.

Reviewer #3:

Remarks to the Author:

We appreciate that the authors have generally been responsive to each of our individual comments, but our prior judgement, but in the absence of substantially new demonstrations of experimental impact, our prior assessment remains the same: "we believe the technology demonstrated here lacks the 'game-changing' impact necessary for publication." We have no doubt that these activator domains function as described, but we are highly skeptical that the field will start to move towards these new TADs en masse, in which case this seems better suited to a more specialized journal.

RESPONSE: We appreciate the constructive feedback from Reviewer #3 throughout the review process of this manuscript. We strongly believe that the newly discovered transactivation domains, compact engineered multipartite TAD modules, and CRISPR-DREAM platforms will be extremely useful for programmable transcription activation. In fact, very recently, two preprint articles: (<https://www.biorxiv.org/content/10.1101/2023.05.12.540558v1.full> and <https://www.biorxiv.org/content/10.1101/2023.06.02.543492v3.full>) also demonstrated the discovery of TAD modules from human and viral proteins using high throughput methods, suggesting that the discovery and use of new and improved TAD modules for programmable transcription modulation remains a high priority among the research community and possesses considerable interest and impact. Therefore, we believe that our engineered TAD modules and CRISPR-DREAM platforms are useful in many aspects including: the novelty of the TADs, our comprehensive rationale engineering approach, our rigorous testing and validation of single and multipartite TADs, and our demonstrations of portability and versatility of these TADs. Further, our demonstration that these new human-derived and compact TADs are well tolerated and efficacious in clinically relevant primary cell types is advantageous and meaningful.

Final Decision Letter:

Dear Isaac,

I am pleased to inform you that your Article, "Compact engineered human mechanosensitive transactivation modules enable potent and versatile synthetic transcriptional control", has now been accepted for publication in Nature Methods. Your paper is tentatively scheduled for publication in our November print issue, and will be published online prior to that. The received and accepted dates will be 26 Jan 2023 and 05 Sep 2023. This note is intended to let you know what to expect from us over the next month or so, and to let you know where to address any further questions.

Over the next few weeks, your paper will be copyedited to ensure that it conforms to Nature Methods style. Once your paper is typeset, you will receive an email with a link to choose the appropriate publishing options for your paper and our Author Services team will be in touch regarding any additional information that may be required.

You will receive a link to your electronic proof via email with a request to make any corrections within 48 hours. If, when you receive your proof, you cannot meet this deadline, please inform us at

rjsproduction@springernature.com immediately.

Please note that *Nature Methods* is a Transformative Journal (TJ). Authors may publish their research with us through the traditional subscription access route or make their paper immediately open access through payment of an article-processing charge (APC). Authors will not be required to make a final decision about access to their article until it has been accepted. [Find out more about Transformative Journals](https://www.springernature.com/gp/open-research/transformative-journals)

Authors may need to take specific actions to achieve [compliance with funder and institutional open access mandates](https://www.springernature.com/gp/open-research/funding/policy-compliance-faqs). If your research is supported by a funder that requires immediate open access (e.g. according to [Plan S principles](https://www.springernature.com/gp/open-research/plan-s-compliance)) then you should select the gold OA route, and we will direct you to the compliant route where possible. For authors selecting the subscription publication route, the journal's standard licensing terms will need to be accepted, including [self-archiving policies](https://www.springernature.com/gp/open-research/policies/journal-policies). Those licensing terms will supersede any other terms that the author or any third party may assert apply to any version of the manuscript.

Your paper will now be copyedited to ensure that it conforms to Nature Methods style. Once proofs are generated, they will be sent to you electronically and you will be asked to send a corrected version within 24 hours. It is extremely important that you let us know now whether you will be difficult to contact over the next month. If this is the case, we ask that you send us the contact information (email, phone and fax) of someone who will be able to check the proofs and deal with any last-minute problems.

If, when you receive your proof, you cannot meet the deadline, please inform us at rjsproduction@springernature.com immediately.

Once your manuscript is typeset and you have completed the appropriate grant of rights, you will receive a link to your electronic proof via email with a request to make any corrections within 48 hours. If, when you receive your proof, you cannot meet this deadline, please inform us at rjsproduction@springernature.com immediately.

Once your paper has been scheduled for online publication, the Nature press office will be in touch to

confirm the details.

Once your paper has been scheduled for online publication, the Nature press office will be in touch to confirm the details.

Content is published online weekly on Mondays and Thursdays, and the embargo is set at 16:00 London time (GMT)/11:00 am US Eastern time (EST) on the day of publication. If you need to know the exact publication date or when the news embargo will be lifted, please contact our press office after you have submitted your proof corrections. Now is the time to inform your Public Relations or Press Office about your paper, as they might be interested in promoting its publication. This will allow them time to prepare an accurate and satisfactory press release. Include your manuscript tracking number NMETH-A51570B and the name of the journal, which they will need when they contact our office.

About one week before your paper is published online, we shall be distributing a press release to news organizations worldwide, which may include details of your work. We are happy for your institution or funding agency to prepare its own press release, but it must mention the embargo date and Nature Methods. Our Press Office will contact you closer to the time of publication, but if you or your Press Office have any inquiries in the meantime, please contact press@nature.com.

Nature Portfolio journals [encourage authors to share their step-by-step experimental protocols](https://www.nature.com/nature-research/editorial-policies/reporting-standards#protocols) on a protocol sharing platform of their choice. Nature Portfolio 's Protocol Exchange is a free-to-use and open resource for protocols; protocols deposited in Protocol Exchange are citable and can be linked from the published article. More details can found at a

href="https://www.nature.com/protocolexchange/about"
target="new">www.nature.com/protocolexchange/about.

Best regards,
Madhura

Madhura Mukhopadhyay, PhD
Senior Editor
Nature Methods